# AMPA-receptor specific biogenesis complexes control synaptic transmission and intellectual ability

Aline Brechet[1,*], Rebecca Buchert[2,†], Jochen Schwenk[1,3,*], Sami Boudkkazi[1,*], Gerd Zolles[1], Karine Siquier-Pernet[4], Irene Schaber[1], Wolfgang Bildl[1], Abdelkrim Saadi[5], Christine Bole-Feysot[4], Patrick Nitschke[4], Andre Reis[2], Heinrich Sticht[6], Nouriya Al-Sanna'a[7], Arndt Rolfs[3,8], Akos Kulik[1,3], Uwe Schulte[1,3,9], Laurence Colleaux[4], Rami Abou Jamra[2,10] & Bernd Fakler[1,3]

AMPA-type glutamate receptors (AMPARs), key elements in excitatory neurotransmission in the brain, are macromolecular complexes whose properties and cellular functions are determined by the co-assembled constituents of their proteome. Here we identify AMPAR complexes that transiently form in the endoplasmic reticulum (ER) and lack the core-subunits typical for AMPARs in the plasma membrane. Central components of these ER AMPARs are the proteome constituents FRRS1l (C9orf4) and CPT1c that specifically and cooperatively bind to the pore-forming GluA1-4 proteins of AMPARs. Bi-allelic mutations in the human *FRRS1L* gene are shown to cause severe intellectual disability with cognitive impairment, speech delay and epileptic activity. Virus-directed deletion or overexpression of FRRS1l strongly impact synaptic transmission in adult rat brain by decreasing or increasing the number of AMPARs in synapses and extra-synaptic sites. Our results provide insight into the early biogenesis of AMPARs and demonstrate its pronounced impact on synaptic transmission and brain function.

[1] Institute of Physiology, Faculty of Medicine, University of Freiburg, Hermann-Herder-Str. 7, Freiburg 79104, Germany. [2] Institute of Human Genetics, University of Erlangen, Schwabachanlage 10, Erlangen 91054, Germany. [3] Center for Biological Signaling Studies (BIOSS), Schänzlestr. 18, Freiburg 79104, Germany. [4] INSERM UMR 1163, Paris-Descartes-Sorbonne Paris Cité University, Institut IMAGINE, Necker-Enfants Malades Hospital, Paris 75015, France. [5] Department de Neurologie, Etablissement Hospitalier Specialisé de Benaknoun, Algers, Algeria. [6] Institute of Biochemistry, Emil-Fischer Center, Fahrstraße 17, Erlangen 91054, Germany. [7] Dharan Health Center, 8131 Medical Access Rd 1, Gharb al Dharan, Dharan 34465, Saudi Arabia. [8] Albrecht-Kossel-Institute for Neuroregeneration, Medical University Rostock, Gehlsheimerstr. 20, Rostock 18147, Germany. [9] Logopharm GmbH, Schlossstr. 14, March-Buchheim 79232, Germany. [10] Institute of Human Genetics, University of Leipzig Hospitals and Clinics, Philipp-Rosenthal-Str. 55, 04103 Leipzig, Germany. * These authors contributed equally to this work. † Present address: Institute of Medical Genetics and Applied Genomics, University of Tübingen, Tübingen 72076, Germany. Correspondence and requests for materials should be addressed to L.C. (email: laurence.colleaux@inserm.fr) or to R.A.J. (email: rami.aboujamra@medizin.uni-leipzig.de) or to B.F. (email: bernd.fakler@physiologie.uni-freiburg.de).

Fast excitatory neurotransmission that is fundamental for the operation of normal brain function mainly relies on AMPA-type glutamate receptors (AMPARs). These ionotropic receptors mediate a large part of the excitatory postsynaptic currents (EPSCs) that drive point-to-point transmission in glutamatergic synapses and control both propagation of the electrical signal and the influx of calcium ions into the postsynaptic spine[1–5]. By these means, AMPARs promote formation and maturation of new synapses and trigger activity-dependent processes that feedback onto the AMPARs thus altering amplitude and properties of the EPSCs[6–11]. In combination, changes in signal transduction and wiring are thought to endow excitatory neurotransmission with the activity-initiated plasticity that underlies learning and memory[12–14].

AMPARs are macromolecular complexes whose functional properties and cell biology are defined by their proteome, the ensemble of their protein building blocks[15,16]. Accordingly, the channel properties such as gating, calcium permeability or block by polyamines are determined by the receptor core that is built from tetramers of the pore-forming GluA1-4 proteins and up to four TARP, CNIH or GSG1l proteins[15–19]. All these core constituents display pronounced spatiotemporal diversity in their expression and distribution, mirroring the marked variations of AMPAR-mediated EPSCs among individual neurons, across brain regions and at distinct states of activity and development[15]. In contrast, the proteome constituents that surround the core subunits, a set of ~20 mostly transmembrane or secreted proteins, may be involved in various aspects of synapse physiology[16]. So far, however, the primary function(s) and subcellular distribution of most of these proteins remain unknown. In particular, the assembly of the receptor complexes or their subsequent processing and trafficking has not been elucidated. Among these peripheral constituents is FRRS1l (ferric-chelate reductase 1-like, also known as C9orf4 refs 15,16), a predicted single-pass transmembrane protein which as yet lacks annotation of any primary function(s) and shows little regional diversity appearing in all brain regions at amounts proportional to that of the GluA tetramers[15].

Here we used reverse proteomics and serial affinity purifications (APs) combined with high-resolution mass spectrometry to uncover particular assemblies within the AMPAR proteome in the rodent brain. We identified a subset of AMPARs that include FRRS1l and CPT1c (carnitine O-palmitoyltransferase 1c) as key subunits and that fundamentally differ from receptor channels at the surface plasma membrane. Subsequent work using genetic analyses, biochemical binding studies, immuno-electron microscopic (EM) and functional recordings showed that FRRS1l-containing AMPAR assemblies represent the priming step of AMPAR biogenesis in endoplasmic reticulum (ER) membranes and strongly influence synaptic transmission. Defective biogenesis by different mutations of FRRS1l underlies severe intellectual disability in humans.

## Results

### Proteomic analyses identify distinct AMPAR assemblies.

For assessment of the cell biology of yet uncharacterized AMPAR subunits, we initially set up reverse proteomic analyses combining APs with antibodies (ABs) that target the non-GluA constituents with high-resolution nano-flow tandem mass spectrometry (nano-LC MS/MS) and protein quantification based on calibrated peptide signals (label-free QconCAT[16,20]). This procedure provides molecular abundance values for all constituents of the AMPAR proteome (Supplementary Data 1) and enables comparison of proteins in different AP samples over a broad

dynamic range of four orders of magnitude. Figure 1a illustrates the result of such initial APs with several anti-FRRS1l, anti-TARP and anti-GluA ABs on membrane preparations from the whole rat brain. Strikingly, the abundance heat map of these APs suggested the existence of two mutually exclusive populations of AMPAR assemblies within the proteome: One population comprising FRRS1l, CPT1c, Sac1 as well as ABHDs 6/12 and PORCN, the other containing the core subunits (CNIHs, TARPs, GSG1l) and the remainders of the peripheral constituents (red frames in Fig. 1a). The only major elements shared by both AMPAR assemblies were the GluA proteins; consequently, anti-GluA APs performed in parallel effectively retained the complete set of constituents of the AMPAR proteome (Fig. 1a).

For more detailed investigation of FRRS1l-containing protein assemblies in the rat brain, we next performed two sets of inverse serial AP experiments (termed 'two-step APs') that are schematized in Fig. 1b (right panel): in the first AP series, the entire complement of FRRS1l protein, co-assembled with AMPARs or AMPAR-free, was extracted from membrane preparations by a mixture of anti-FRRS1l ABs prior to a target-depleting anti-GluA1-4 AP, and in the second AP series, FRRS1l was entirely affinity-isolated from membrane preparations that have been depleted of all AMPARs via an AP with anti-GluA1-4 ABs (Fig. 1b, right panel, Supplementary Fig. 1a). Quantitative evaluation of the protein amounts determined from nano-LC MS analysis of these serial APs thus provided direct information on (i) the assembly of any proteome constituent with FRRS1l-containing versus FRRS1l-free AMPARs (Fig. 1b, left panel, bars in red and blue, respectively) and (ii) for co-assembly of any proteome constituent with the FRRS1l protein independent of GluA1-4 (Fig. 1b, right panel, brown bars). The respective results led to the following major observations: First, FRRS1l assembles into an average 15–20% of all AMPARs (at steady state) as indicated by the relative amounts of GluA1-4 proteins in the anti-FRRS1l AP (red bars for GluA1-4, Fig. 1b, left two-step AP). Second, 80–85% of AMPARs lack FRRS1l but instead contain the core constituents TARPs, CNIHs and GSG1l (blue bars for GluA1-4 and the core subunits, Fig. 1b, left two-step AP). Third, FRRS1l effectively associates with CPT1c and Sac1 independent of GluA1-4-containing complexes (Fig. 1b, brown bars, right two-step AP), while robust co-assembly with ABHDs 6/12 and PORCN requires the pore-forming GluA1-4 subunits (Fig. 1b, red bars in the left two-step AP). Fourth, the secreted/extracellular constituents of the AMPAR proteome, Noelins1-3, Brorin and Brorin-2l, as well as Neuritin and LRRT4 were exclusively found as subunits of FRRS1l-free AMPARs (both two-step APs). These results were recapitulated in two-step APs using membrane fractions from the entire mouse brain as a source material (Supplementary Fig. 1b).

### FRRS1l and CPT1c are AMPAR-selective interactors.

In addition to the analyses within the AMPAR proteome, we aimed at a more comprehensive look into interaction partners of FRRS1l by means of APs using four different anti-FRRS1l ABs and two different ABs targeting CPT1c on membrane fractions from the whole rat brain (see Methods section). Preimmunization immunoglobulins G (IgG) as well as FRRS1l-depleted membrane fractions and membrane fractions from CPT1c knockout animals served as negative controls defining specificity of (co)-purification (refs 16,21,22; see Methods section). All proteins exceeding the specificity thresholds in at least two different anti-FRRS1l or anti-CPT1c APs were dubbed bona fide interaction partners of the respective target and summarized in Fig. 2a together with their abundance values (obtained across the different APs). These abundance plots showed that both FRRS1l and CPT1c mutually

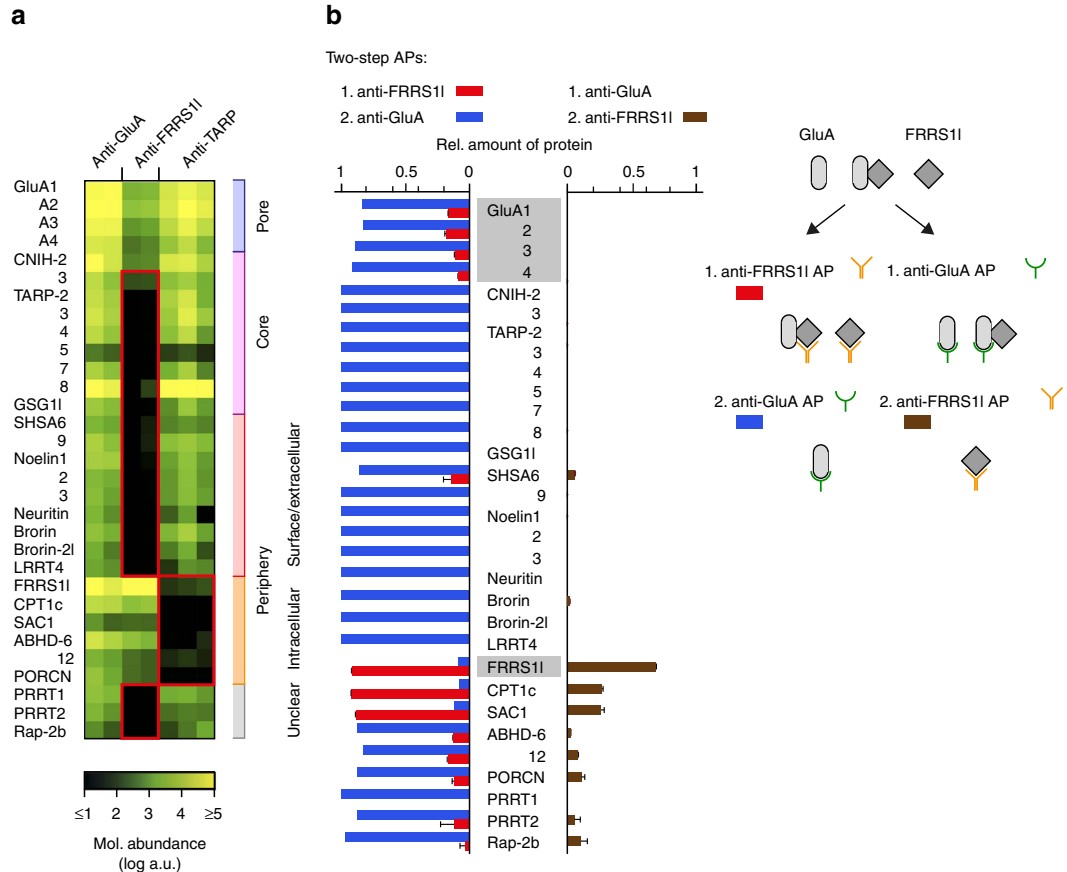

**Figure 1 | Identification of distinct populations of AMPAR assemblies in the rat brain.** (**a**) Heat map indicating the molar abundance of AMPAR constituents determined in APs with various ABs targeting GluA1–4 (mixture of anti-GluA ABs), FRRS1l (anti-FRRS1l-a, anti-FRRS1l-b) and TARPs 2, 3, 4, 8 (anti-TARP-a, anti-TARP-b, anti-TARP-c) in membrane fractions from total rat brain solubilized with CL-47. Note the distinct subgroups of constituents co-purified with the pore-forming GluA1–4 proteins in anti-FRRS1l and anti-TARP APs highlighted by red boxes. Annotations on the right reflect molecular architecture and/or subcellular localization reported in literature or public databases. (**b**) Relative amounts of AMPAR constituents determined in two-step APs schematized on the right from CL-47 solubilized membrane fractions of total adult rat brains. Bars in red and blue depict the amount of AMPAR constituents in a target-depleting anti-FRRS1l AP (red bars, mean ± s.d. of three measurements) and a subsequent target-depleting anti-GluA AP (blue), each determined as fraction of its summed protein amounts in the two APs. Brown bars illustrate relative amounts of AMPAR constituents (mean ± s.d. of three measurements) in an anti-FRRS1l AP using the flow through of a target-depleting anti-GluA AP as input divided by the summed protein amount determined for any constituent in both APs from rat membranes. Note co-assembly of FRRS1l with GluA1–4 and only a subset of AMPAR proteome constituents.

co-purified each other at amounts emphasizing their effective co-assembly (close to stoichiometric in anti-FRRS1l APs). Moreover, both target proteins effectively co-purified the same subset of AMPAR proteome constituents, the pore-forming GluA1–4, Sac1, ABHDs 6/12 and PORCN (Fig. 2a). Noteworthy, the abundances of these AMPAR subunits were similar among each other in either target AP and the amounts of co-purified AMPARs (defined as sum of GluA1–4 abundances/4; [15]) exceeded by far the values obtained for any of the remaining interactors identified for FRRS1l and CPT1c. The latter are all indicated in public databases (UniProtKB/Swiss-Prot, EMBL-EBI) being membrane proteins with different topology and suggested preferred localization to the ER (amounts <0.5% of AMPARs; Fig. 2a).

These results unraveled an almost exclusive partnership of FRRS1l and CPT1c with AMPARs and show that both proteins co-associate into the same AMPAR assemblies. This view was further corroborated by direct comparison between the entire pools of AMPARs affinity-isolated from wild-type (WT) and CPT1c knockout animals[23]. As shown in Fig. 2b, deletion of CPT1c resulted in a markedly reduced assembly of FRRS1l (>90%) and Sac1 (>70%) into AMPAR complexes.

Interestingly, the amounts of several other AMPAR constituents appeared altered by roughly between 10% and 50% (Fig. 2b).

Together, the results of our proteomic analyses established complexes assembled from GluA1–4, FRRS1l, CPT1c and Sac1 as particular AMPAR assemblies that largely differ from the well-known AMPARs in the plasma membrane by lacking the entire set of core constituents. Instead, as CPT1c and Sac1 have been reported as ER proteins[23,24], the newly identified population of AMPARs may be localized to this intracellular membrane compartment—provoking the question of its role in AMPAR physiology.

**Mutations in the *FRRS1L* gene cause intellectual disability.** In the course of two independent systematic studies analysing cohorts of consanguineous families for genes causative for autosomal recessive intellectual disability, we carried out whole-exome sequencing (WES) followed by appropriate filtering (refs 25,26, see Methods section) on patients and unaffected family members (WES data and identified genes were communicated on the CARID (Consortium of Autosomal Recessive Intellectual Disability) platform). In three independent families from Algeria,

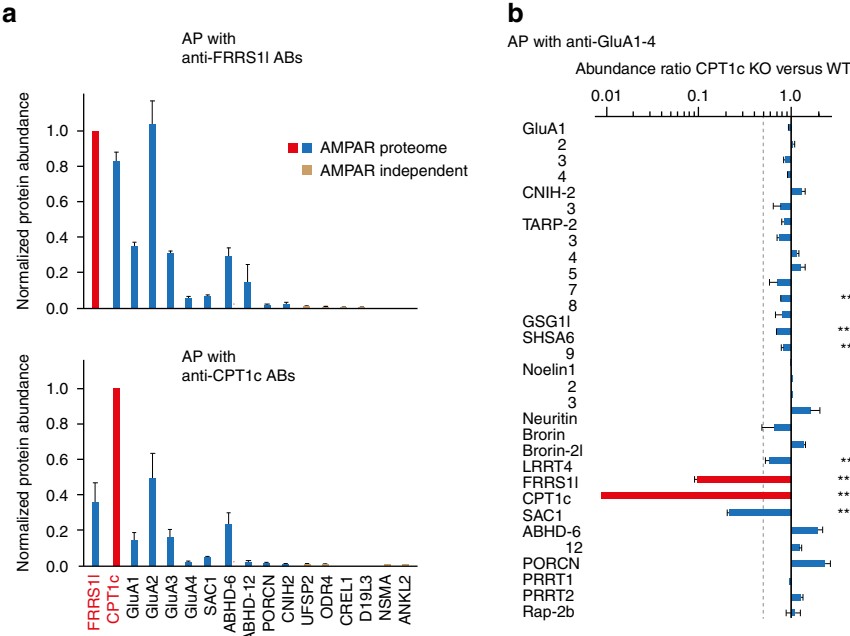

**Figure 2 | Specific co-assembly of FRRS1l and CPT1c into AMPAR complexes with distinct subunit composition.** (**a**) Amounts of proteins specifically co-purified in APs with four different anti-FRRS1l ABs (upper panel) and two different anti-CPT1c ABs (lower panel) normalized to the amount of purified FRRS1l or CPT1c, respectively (red bar). Data are mean ± s.d. of five (four ABs, one AB mixture, see Methods section) and three (two ABs, one AB mixture) independent APs, respectively. (**b**) Protein abundance ratios (mean ± s.e.m. of four measurements) determined for the indicated proteome constituents in anti-GluA1-4 APs from brain membrane fractions prepared from both WT and CPT1c knockout mice (normalized to GluA1-4). Note the pronounced decrease of CPT1c, FRRS1l and Sac1 in AMPARs from knockout animals. Dashed line indicating 50% reduction in protein amount is shown for orientation; asterisks denote statistical significance for protein amounts being different between WT and KO (**, ***P values of 0.01 and 0.001 for Students' t-test, respectively).

Syria and Saudi Arabia (Fig. 3a, Supplementary Table 1), these analyses revealed three homozygous variants in the *FRRS1L* gene segregating with the disorder and fitting the recessive mode of inheritance suggested by the pedigrees (Fig. 3a): a missense variant leading to a lysine-to-glutamate exchange at residue 155 (NM_014334.3: c.463A > G; p.K155E) in family A, a one-base-pair deletion causing a frameshift and premature stop (NM_014334.2.3: c.584delT; p.V195E fs*35) in family B and a nonsense mutation resulting in a premature stop at residue 321 (NM_014334.3: c.961C > T, p.Q321*) in family C (Fig. 3a,b). Filtering WES data for compound heterozygous variants, X-linked or heterozygous *de novo* variants did not provide any additional candidate gene (Supplementary Table 1). Capillary sequencing further confirmed the *FRRS1L* variants and their segregation in all families (Supplementary Fig. 2).

The identified alterations in the *FRRS1L* gene led to a similar phenotype in all patients, albeit differences in severity and disease course were observed with the individual mutations. Thus, after unremarkable pregnancy and birth and an uncomplicated neonatal period (with normal growth parameters), all patients showed a pronounced delay or failure in reaching developmental milestones, moderate-to-severe intellectual disability, delayed or strongly restricted speech development and seizures (Fig. 3c and Supplementary Table 2). Computed tomographic scans or magnetic resonance imaging from the brain did not reveal any clear abnormalities, and growth parameters, including head circumference, height and weight, were normal or slightly beyond the normal range (Supplementary Table 2). In addition, all patients of families B and C showed muscular hypotonia and, at an age of about 12 months, exhibited regression finally leading to loss of all acquired skills. Both neuroregression and muscular hypotonia were not observed with the patients in family A, making the phenotype segregating with the K155E mutation

appear less severe. In addition, the affected siblings in family A achieved independent walking, which all patients of families B and C failed (Fig. 3, Supplementary Table 2).

Homozygous mutations in the *FRSS1L* gene have been very recently reported to cause epileptic encephalopathy[27,28]; interestingly, the p.Q321* variant was observed in two distinct additional Saudi Arabian families with a similar clinical presentation, suggesting a founder mutation in this population.

Together with these latest reports, our results show that bi-allelic mutations in *FRRS1L* cause autosomal-recessive intellectual disability and suggest the importance of FRRS1l-containing AMPARs for normal brain development and function.

**FRRS1l–CPT1c-containing AMPARs are restricted to the ER.** To gain insight into the cell biology of FRRS1l, we next went to its heterologous expression, either alone or in combination with the identified partners of the AMPAR proteome, and used confocal fluorescence microscopy, biochemistry and electrophysiology as complementary techniques for analysis (Fig. 4, Supplementary Figs 3–5).

First, we probed complex formation of FRRS1l, CPT1c and Sac1 with GluA1/2 by combinatorial expression and subsequent affinity isolation of the resulting AMPAR assemblies via anti-GluA1/2 or anti-FRRS1l ABs. Western blot-probed gel separations of anti-GluA APs revealed that FRRS1l and CPT1c were efficiently co-purified with GluA1/A2 when present together (Fig. 4a), while individual co-purification of either protein was markedly less abundant (Fig. 4a). Similarly, Sac1 displayed weak or no interaction with GluA1/A2 alone or together with either FRRS1l or CPT1c but was co-purified in anti-GluA and anti-FRRS1l APs when FRRS1l and CPT1c complexes were

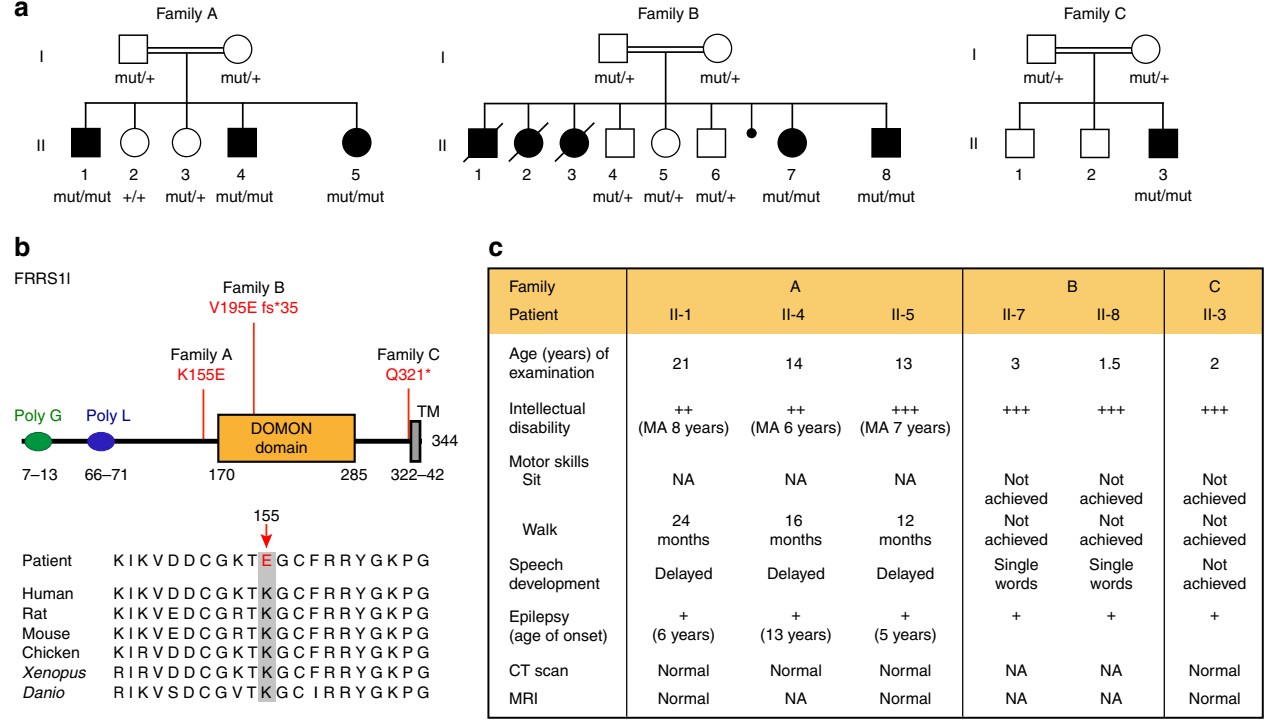

**Figure 3 | Mutations in FRRS1l identified as disease causing in patients with severe intellectual disability.** (**a**) Pedigree (and genotypes) of the three different families detailed in the text. Affected patients are indicated by filled symbol; slashes refer to deceased individuals. (**b**) Schematic representation of the FRRS1l protein (as given in the UniProtKB/Swiss-Prot database) together with the alterations identified in the indicated families and alignment of the primary sequence stretch around lysine 155 (K155) across species. TM is a predicted transmembrane domain. (**c**) Summary of the clinical features observed in the patients from families A–C. MA denotes mental age, NA means not investigated.

present for co-assembly (Fig. 4a). In addition, the anti-FRRS1l AP demonstrated robust formation of ternary complexes between FRRS1l, CPT1c and Sac1 (Fig. 4a). Thus, in co-assembled form, FRRS1l and CPT1c promote (or stabilize) their mutual interaction with the AMPAR pore and serve as binding platform for Sac1, in close agreement with the results obtained before from the rat brain (Figs 1 and 2).

Further analysis, by confocal microscopy, revealed another profound effect of FRRS1l–CPT1c interaction: while sole FRRS1l protein was predominantly detected at the plasma membrane (Fig. 4b, upper left panel), CPT1c-assembled FRRS1l appeared entirely re-distributed to intracellular membrane compartments where its staining largely overlapped with those of CPT1c (Fig. 4b, right panels) and the ER-marker calnexin (Supplementary Fig. 3a) or edited GluA2 (Supplementary Fig. 3b). This CPT1c-mediated re-distribution of FRRS1l was also observed with GluA-associated FRRS1l as revealed in a surface biotinylation assay (see Methods section) and by current recordings from outside-out patches: In either approach, FRRS1l was no longer detected in the plasma membrane upon co-expression of CPT1c (Supplementary Fig. 4a,b). Importantly, the redistribution of FRRS1l was specific for the ER-resident CPT1c, as replacing CPT1c with CPT1a, an enzymatically active member of the carnitine O-palmitoyltransferase family located in the outer mitochondrial membrane[29], failed to promote re-distribution of FRRS1l (that is, targeting to the plasma membrane was unaffected; Fig. 4b, middle and lower left panel; Supplementary Fig. 4b).

Next, we looked closer into the assembly of FRRS1l with GluA proteins in the ER and into ER-based processing of FRRS1l. For assembly analysis, we used native gel electrophoresis from brain membrane fractions and anti-FRRS1l APs from cells co-expressing GluAs1-4 individually with FRRS1l and CPT1c.

As illustrated in Supplementary Fig. 5, western blot analysis of APs and native polyacrylamide gel electrophoresis (BN-PAGE) showed that FRRS1l/CPT1c do not exhibit any subtype preference in their assembly with GluA1-4 (Supplementary Fig. 5a) but exclusively co-assemble with the GluAs into high molecular weight (MW) complexes (Supplementary Fig. 5b). Quantitative analysis by cryo-slicing BN-MS[30] in fact showed that these assemblies are formed by about equimolar ratios of GluAs and FRRS1l/CPT1c in line with roughly four FRRS1l/CPT1c complexes co-assembled into each GluA tetramer (Supplementary Fig. 5b).

ER-based processing of FRRS1l was investigated as western blot probing SDS–PAGE separations of brain membranes with anti-FRRS1l revealed two bands with distinct MW (Fig. 4c, left panel). Detailed MS-based sequence analysis of native and recombinant FRRS1l showed that the higher MW band corresponds to the full-length protein (Supplementary Fig. 6a), while the lower MW band is the protein truncated at residue serine 317 (S317, Fig. 4c, Supplementary Fig. 6a). In addition, mass spectrometry identified S317 as an attachment site for a glycosylphosphatidylinositol (GPI)-anchor through the typical ethanolamine moiety (refs 31,32, Fig. 4c), in line with suggestions from prediction algorithms[33]. During GPI-anchoring, proteolytic cleavage and side-chain modification are mediated by a GPI transamidase that uses S317 as C-terminal α-site and the hydrophobic domain at the C-terminus of FRRS1l (Fig. 4c, Supplementary Fig. 6b) as a necessary structural cofactor[32]. Accordingly, FRRS1l is anchored in the ER membrane either via the lipid moiety of the GPI-anchor or via the hydrophobic domain in the C-terminus[34]. In any case, the particular importance of membrane anchoring for interaction of FRRS1l with the AMPAR pore became evident from further mutagenesis: Deletion of the C-terminus at alanine 318 (Δ318) entirely

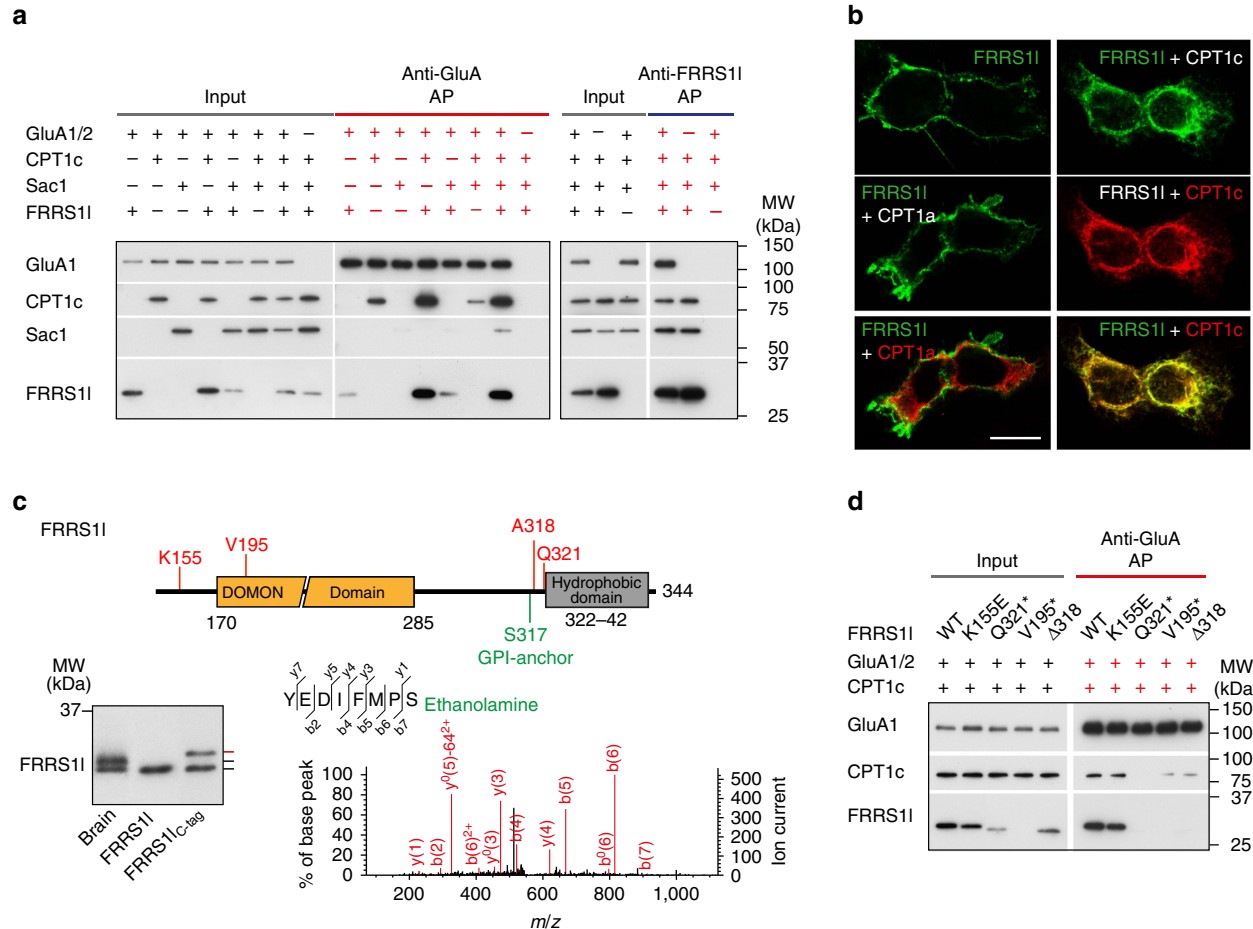

**Figure 4 | Assembly of WT and mutant FRRS1l with CPT1c and significance for ER localization and association with GluA proteins.** (**a**) Input and eluates of anti-GluA1 and anti-FRRS1l APs separated by SDS–PAGE and western blot probed for the indicated proteins. Note strong cooperativity of FRRS1l and CPT1c binding to AMPARs. (**b**) Representative confocal fluorescence images of tsA-201 cells expressing FRRS1l alone (upper left) or together with CPT1a (middle and lower left) or CPT1c (right); fluorescence staining as indicated by the colour coding (with anti-FRRS1l-a, anti-CPT1a or anti-CPT1c). Scale bar is 10 μm. Note marked re-distribution of FRRS1l upon co-expression of CPT1c from plasma membrane to intracellular membranes (FRRS1l staining: 93% (of 501 cells co-expressing FRRS1l and CPT1c) intracellular only, 6.8% intracellular and plasma membrane, 0.2% plasma membrane only). (**c**) Left: SDS–PAGE separation of membrane preparations from brain and culture cells (expressing the indicated proteins) western blot probed with anti-FRRS1l-a. Marks on the right refer to the distinct MW bands, red denotes FRRS1l with additional mass introduced by the HA-6His tag (Supplementary Fig. 6). Right: CID MS/MS spectrum of the indicated FRRS1l peptide carrying an ethanolamine moiety at αS317 as a specific marker for GPI-anchoring (Supplementary Fig. 6). Inset: Scheme highlighting the hydrophobic domain and relevant residues in the C-terminus of FRRS1l. (**d**) Input and eluates of anti-GluA1 APs separated by SDS–PAGE and western blot probed for the indicated proteins. Note the reduction or failure of the mutant FRRS1l proteins to assemble with the GluA proteins.

abolished the robust co-assembly observed with WT FRRS1l (Fig. 4d).

**Disease mutations disturb FRRS1l-AMPAR interaction.** Finally, we investigated the mutations identified in patients with intellectual disability for their impact on protein expression and assembly of FRRS1l. As shown in Fig. 4d, both truncation mutations, FRRS1l(Q321*) and FRRS1l(V195E fs*35), failed co-assembly with GluA1/A2 (Fig. 4d) and also appeared less abundantly expressed (or less stable) than the WT protein. In contrast, the K155E substitution mutant was still able to interact with GluA1/A2, albeit less effectively than WT FRRS1l (Fig. 4d).

Together, the results obtained from heterologous expression in culture cells demonstrated robust complex formation of FRRS1l and CPT1c, as well as their co-assembly with GluA tetramers in the ER. This co-assembly with the AMPAR pore and/or the stability of the FRRS1l protein is largely affected by the mutations identified in patients with intellectual disability.

**FRRS1l-containing AMPARs in brain neurons localize to the ER.** The subcellular distribution of FRRS1l seen in transfected culture cells was entirely recapitulated in hippocampal neurons as detected by confocal microscopy and pre-embedding immunogold EM in respective slice preparations of the rat brain. Thus immuno-fluorescence staining of FRRS1l closely overlapped with that of the ER-marker calnexin (Fig. 5a), and gold particles coupled to anti-FRRS1l were almost exclusively detected at ER membranes, but not over the Golgi complex, and only very rarely at/or close to the plasma membrane (Fig. 5b). In more detail, 96.1% of all immunoparticles evaluated in 29 neurons (815 out of a total of 848 particles) showed ER localization, while the remaining 3.9% (33 particles) appeared at or next to the plasma membrane. In either case, the staining was mostly observed in the soma or the proximal dendrites but did not appear in the synaptic compartment (Fig. 5a).

**FRRS1l/CPT1c assemblies impact synaptic transmission.** For investigating the functional significance of FRRS1l–CPT1c-

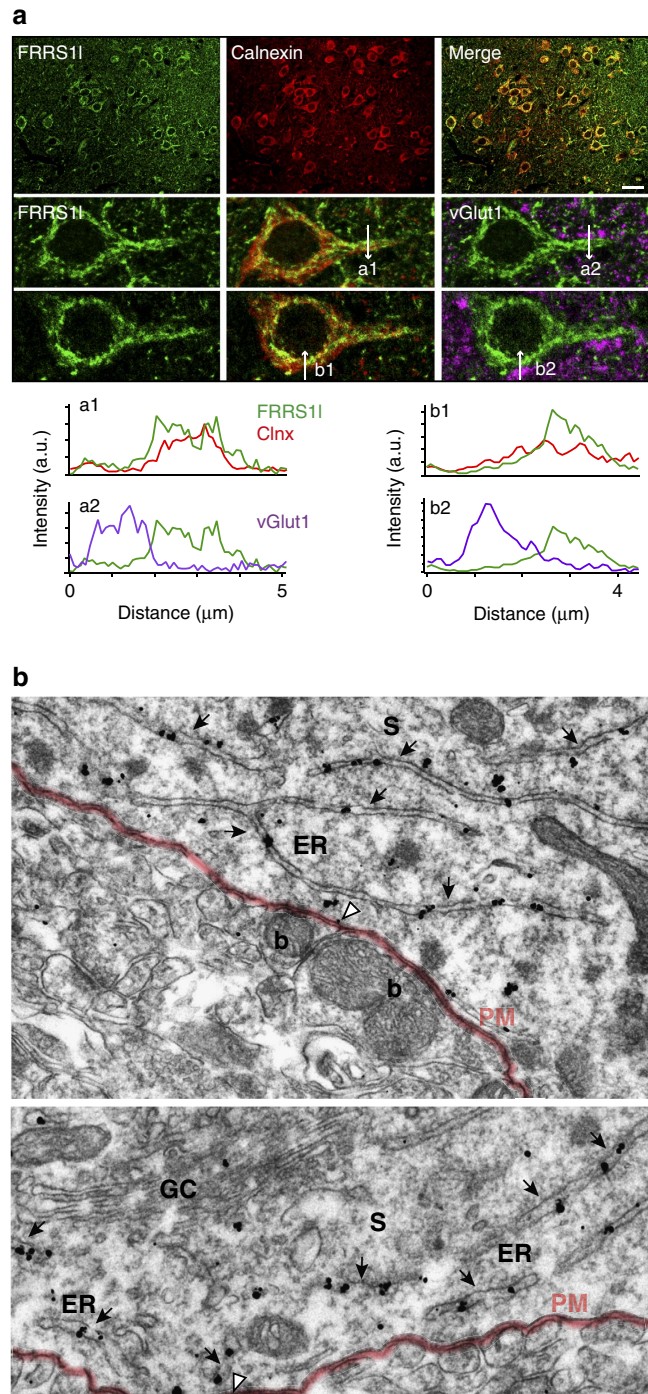

**Figure 5 | Subcellular localization of FRRS1l in neurons of the hippocampus.** (**a**) Confocal images of the hilar region of hippocampal slices prepared from adult rats and selected neurons therein triple stained with anti-FRRS1l, the ER-marker anti-Calnexin and the synapse marker anti-vGlut1. Graphs in the lower panel are fluorescence intensity scans of the annotated markers measured along the indicated lines on the confocal images of the selected neurons. (**b**) Electron micrographs showing immuno-reactivity for FRRS1l in somata (S) of CA1 pyramidal cells as detected by the pre-embedding immunogold method. Immunoparticles labelling FRRS1l were abundant over the endoplasmic reticulum (ER, arrows) but not on membranes of the Golgi complex (GC in lower panel). Particles (open arrowheads) occasionally appeared close to or at the somatic plasma membrane (PM, red line) targeted by boutons (b) of putative GABAergic neurons. Scale bar: 200 nm.

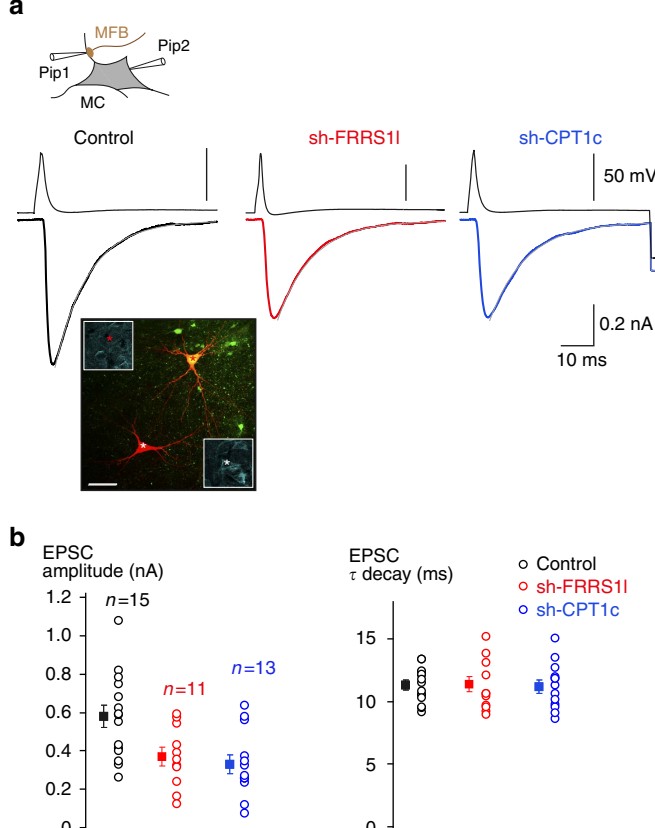

**Figure 6 | Alterations in EPSCs upon knockdown of FRRS1l in individual MFB–MC synapses.** (**a**) Representative action potential and EPSC traces determined by paired bouton recordings (upper inset) in hippocampal slices from MFB–MC synapses of an uninfected MC (control, left) or MCs transduced with sh-FRRS1l (middle) or sh-CPT1c (right). Current and time scaling as indicated. Grey lines are mono-exponential fits to the decay phase yielding time constants of 10.4 ms (control), 10.1 ms (sh-FRRS1l) and 10.1 ms (sh-CPT1c). Inset: Confocal fluorescence image of an uninfected (lower) and a sh-FRRS1l (fluorescence of GFP marker) transduced (upper) MC used for recordings (filled with biocytin, red fluorescence); framed images depict anti-FRRS1l staining of the two cells. Scale bar is 10 μm. (**b**) Summary plots of amplitudes (left) and decay time constants of the EPSCs determined in experiments as in **a**. Squares represent mean ± s.e.m. of the experiments shown.

containing AMPARs, we used patch-clamp recordings from various types of hippocampal neurons in brain slices combined with manipulation of protein levels for FRRS1l or CPT1c. The latter was achieved with lentiviruses that were stereotactically administered to P6/7 rats (ref. 35, see Methods section) to promote either knockdown of protein expression via target-specific short hairpin RNAs (sh-FRRS1l, sh-CPT1c, Fig. 6a, for efficiency and specificity of protein knockdown: lower inset and Supplementary Fig. 7) or (over)expression of exogenous protein. Figure 6a illustrates EPSCs mediated by AMPARs in single synapses of hilar neurons in the hippocampus under control conditions and after virus-directed manipulations. The respective EPSCs were measured by the paired-bouton recording technique in postsynaptic mossy cells (MCs) in response to single action potentials elicited in presynaptic mossy fibre boutons (MFBs) by brief current pulses (Fig. 6a, upper inset[35]). Under control conditions (untransduced MCs), the MFB–MC EPSCs that are predominantly carried by CNIH2-containing GluA1/A2 AMPARs[35] exhibited large amplitudes with a mean value

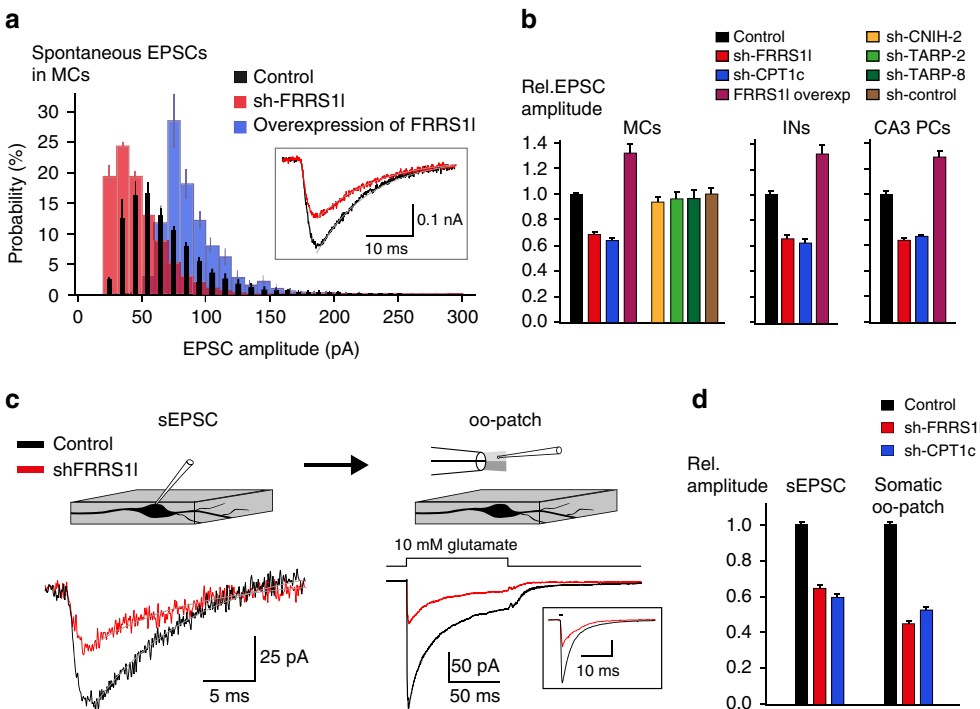

**Figure 7 | Decrease or increase of currents mediated by synaptic and extra-synaptic AMPARs upon knockdown and exogenous (over)expression of FRRS1l in distinct types of hippocampal neurons.** (**a**) Amplitude histograms determined from spontaneous EPSCs recorded in uninfected control or sh-FRRS1l-transduced MCs and in MCs with virally driven overexpression of FRRS1l (at − 70 mV). Data are mean ± s.e.m. of 129 (control), 68 (sh-FRRS1l) and 10 (overexpression) MCs. Inset: EPSCs from a control and a sh-FRRS1l-transduced MC. Current and time scaling as indicated. (**b**) Summary plot of EPSC amplitudes determined in MCs, interneurons (INs) and CA3 pyramidal cells (CA3 PCs) under control conditions (uninfected) or after transduction with the sh-RNAs indicated on the upper right. Data are mean ± s.e.m. of 7–135 MCs, 11–77 INs and 18–61 CA3 PCs; EPSC amplitudes were normalized to the mean amplitude of the control cells. Note that reduction and increase in EPSC amplitude induced by knockdown and overexpression of FRRS1l occurred in all three types of hippocampal neurons. (**c**) Representative spontaneous EPSCs (left) and glutamate-evoked currents in a somatic outside-out (oo) patch both recorded successively either in an uninfected MC (control, black traces) or an MC transduced with sh-FRRS1l (red traces) in a hippocampal slice preparation. Current and time scaling as indicated. (**d**) Summary plot of relative amplitudes obtained for EPSCs and glutamate-evoked AMPAR currents in control MCs and MCs transduced with the indicated sh-RNAs. Data are mean ± s.e.m. of 8 MCs for each condition.

( ± s.e.m.) of 578.1 ± 57.2 pA ($n = 15$; Fig. 6a,b). Virus-driven knockdown of either FRRS1l or CPT1c, however, significantly reduced the amplitude to about 60% (Fig. 6a,b; values of 370.3 ± 47.7 pA ($n = 11$) and 331.6 ± 47.3 pA ($n = 13$) for sh-FRRS1l and sh-CPT1c, respectively; $P < 0.01$, Mann–Whitney U-test), while leaving the time course of the current decay unaffected (values for $\tau_{decay}$ of 11.3 ± 0.4, 11.4 ± 0.6 and 11.2 ± 0.5 ms for control, sh-FRRS1l and sh-CPT1c, respectively). Moreover, the paired-pulse ratios (at 50 Hz) determined in all MFB–MC pairs were similar in untransduced controls and virus-infected MCs (mean ( ± s.e.m.) of 146.0 ± 6.6% ($n = 15$, control), 147.2 ± 7.4% ($n = 11$, sh-FRRS1l) and 145.1 ± 6.0% ($n = 13$, sh-CPT1c)), indicating that knockdown of FRRS1l and CPT1c did not affect the probability of transmitter release from the presynapse.

Similar results as for single MFB–MC synapses were obtained from EPSCs evoked by spontaneous action potentials in the large number of synapses forming onto individual MCs. As demonstrated by distribution histograms, sh-FRRS1l shifted the EPSC amplitudes towards smaller values, while overexpression of FRRS1l resulted in a shift in the opposite direction (Fig. 7a). When compared to untransduced controls, sh-FRRS1l and sh-CPT1c reduced the EPSC amplitudes to values of 0.68 and 0.64, respectively, while exogenous FRRS1l expression increased the EPSC amplitude to 1.32 (Fig. 7b; $P < 0.01$, Mann–Whitney U-test). In neither case was the time course of

the EPSCs affected by these manipulations (values for $\tau_{decay}$ (mean ± s.e.m.) of 11.2 ± 0.2, 10.4 ± 0.3, 11.1 ± 0.3 and 10.7 ± 0.5 ms for control, sh-FRRS1l, sh-CPT1c and FRRS1l overexpression, respectively). Determination of miniature EPSCs in control MCs and MCs transduced with sh-FRRS1l or sh-CPT1c corroborated the amplitude effects (respective values (mean ± s.e.m.) of 22.7 ± 1.8 pA ($n = 5$), 17.8 ± 1.4 pA ($n = 4$) and 18.1 ± 0.3 pA ($n = 4$) for control, sh-FRRS1l and sh-CPT1c, respectively) and suggested that the observed reduction in EPSC amplitude results from a reduced number of functional AMPARs in the postsynapse.

The specificity of the observed knockdown effects on AMPARs was probed in two further sets of experiments: First, by applying either target-unrelated 'control' sh-RNAs or additional sh-RNAs directed against alternative sequences on the FRRS1l or CPT1c mRNAs, and second, by comparing the EPSC components mediated by AMPARs and NMDA (N-methyl-D-aspartate)-type glutamate receptors (NMDARs) in sh-RNA transduced and control MCs. As shown in Fig. 7b, no alterations in the EPSC amplitude were observed in MCs virally transduced with sh-RNAs targeting CNIH2 or with a sh-RNA not recognizing any gene transcript in line with earlier results[35]. Similarly, sh-RNAs targeting TARPs 2 and 8 that were not detected in MCs[35] did not change the EPSC amplitude. In contrast, transduction of MCs with the additional sh-RNAs directed against FRRS1l (sh-FRRS1l-b, -c) or CPT1c (CPT1c-b) resulted in a decrease in EPSC

amplitude (Supplementary Fig. 8) similar to that observed before (Fig. 7b). The AMPA/NMDA ratio determined for the dual component EPSCs recorded at 40 mV (Supplementary Fig. 9) were reduced by approximately 45% in MCs transduced with sh-FRRS1l or sh-CPT1c compared to uninfected control MCs, indicating a selective effect of the protein knockdown on AMPARs (AMPA/NMDA ratio: $1.12 \pm 0.07$, $n = 9$ for control MCs, $0.57 \pm 0.04$, $n = 10$ and $0.65 \pm 0.07$, $n = 9$ for sh-FRRS1l- and sh-CPT1c-transduced MCs, respectively).

Next, we probed the effects of sh-RNA-mediated knockdown of FRRS1l and CPT1c and of FRRS1l (over)expression in other types of hippocampal neurons and investigated their significance for AMPARs in extra-synaptic localization. As illustrated in Fig. 7b, the decrease and increase in EPSC amplitude observed with sh-FRRS1l/sh-CPT1c and FRRS1l (over)expression was not restricted to MCs but was similarly observed in interneurons of the hilar region, as well as in CA3 pyramidal cells (Fig. 7b, middle and right panels, Supplementary Fig. 8). The impact of FRRS1l-/CPT1c knockdown on extra-synaptic AMPARs was tested by dual recordings in MCs as schematized in Fig. 7c (inset): After recording spontaneous EPSCs in whole-cell configuration, the patch-pipette placed on the cell soma was excised into outside-out configuration for fast agonist-application experiments measuring currents through the somatic AMPARs. Representative current traces of such experiments recorded in the same MC together with the summary bar graphs illustrate a concomitant reduction by 40–50% of both EPSCs and somatic AMPAR currents as a consequence of protein knockdown by sh-FRRS1l and sh-CPT1c (Fig. 7c,d).

Together, these results indicated that FRRS1l–CPT1c complexes, although strictly localized to intracellular ER membranes, are able to profoundly affect the amplitude of the EPSCs by controlling the number of AMPARs in both synapses and extra-synaptic sites of the plasma membrane.

## Discussion

We identified FRRS1l and CPT1c as key components of distinct assemblies of AMPARs that are exclusively localized to ER membranes and lack the inner core subunits characteristic for AMPAR complexes at the plasma membrane. These assemblies likely represent an early step in AMPAR biogenesis and markedly affect the amplitude of AMPAR-mediated fast EPSCs. Their significance for brain function is reflected by the loss-of-function mutations in the *FRRS1lL* gene, leading to a severe neurological phenotype with intellectual disability and epilepsy.

For unbiased analysis of complex assemblies encoded by the AMPAR proteome, we used a reverse proteomic approach with non-GluA ABs and (absolute) protein quantification based on high-resolution MS and calibrated peptide signals (Fig. 1a). This approach revealed two subsets of constituents that are mutually exclusive for large parts except for the pore-forming GluA proteins and established a population of AMPAR assemblies with FRRS1l and CPT1c as key components. Subsequent work using two-step and multi-epitope APs (Figs 1b and 2), biochemistry as well as EM and fluorescence microscopy established several key features of FRRS1l–CPT1c-containing AMPARs (Figs 2–5): They (i) lack the known core subunits TARPs, CNIHs and GSG1l, (ii) represent roughly 15–20% of all (steady state) AMPAR assemblies in the rodent brain, (iii) they are exclusively localized to ER membranes through the ER-resident CPT1c, and (iv) they do not exhibit region specificity in distribution across regions of the brain[15]. Important to note, both FRRS1l and CPT1c display pronounced selectivity for assembly with AMPARs (over very few other mostly ER-based proteins) as indicated by our comprehensive proteomic analysis (Fig. 2a). This selectivity contrasts the interactions reported for other ER-localized constituents of the AMPAR proteome, including ABHDs and PORCN[36,37].

The significance of FRRS1l–CPT1c-containing AMPAR assemblies for the cell physiology became evident in functional recordings from various types of hippocampal neurons (Figs 6 and 7). Knockdown of FRRS1l and CPT1c (Figs 6 and 7) and exogenous (over)expression of FRRS1l (Fig. 7) caused a profound decrease or increase of EPSC amplitudes, respectively, by altering the number of AMPARs in the surface membrane of synaptic and extra-synaptic sites. Similarly, decreased mEPSC amplitudes were also reported for cultured hippocampal neurons from CPT1c knockout animals as a consequence of posttranscriptional action(s) of CPT1c (different from de-palmitoylation) impacting surface expression of AMPARs[38,39].

In fact, these results are complemented by two additional (interesting) observations derived from QconCAT-based protein quantification and correlation analyses: first, the protein amounts of AMPAR core subunits, typical for the receptors at the plasma membrane, are reduced proportionally to the amount of FRRS1l in experiments with several independent sh-RNAs (Fig. 8a, left panel), and second, correlation of FRRS1l amounts was maximal with the sum of protein amounts determined for the core subunits (rather than with the amounts of the individual subunits or total AMPARs; Fig. 8a, right panel). Both observations together with their mutual exclusiveness (Fig. 1) strongly suggest that all surface AMPARs (containing TARPs, CNIHs and/or GSG1l) emerge from FRRS1l–CPT1c-containing AMPAR assemblies in the ER.

In the context of a neuron, our results may be combined into a scheme (Fig. 8b), where FRRS1l–CPT1c assemblies act as a general 'catalyst' of AMPAR biogenesis. As a first step in AMPAR biogenesis, newly synthesized GluA tetramers (GluA$_{tetra}$, Fig. 8b) co-assemble with (up to four) FRRS1l–CPT1c complexes (Fig. 4, Supplementary Fig. 5) thus 'priming' them for interaction with the auxiliary proteins CNIHs, TARPs or GSG1l. Binding of these subunits to GluA$_{tetra}$-FRRS1l/CPT1c generates a short-lived transitional complex ('priming of assembly' in Fig. 8b) that has two main consequences: First, it leads to rapid dissociation of FRRS1l/CPT1c, and, second, it establishes stable heteromultimers of GluA and CNIH/TARP subunits. While these stable AMPAR complexes proceed towards the ER-exit sites and are finally delivered to the plasma membrane (through the secretory pathway), FRRS1l/CPT1c complexes remain in the ER, ready for re-association with new GluA$_{tetra}$ and thus re-entering the catalysis cycle. In this model, FRRS1l/CPT1c complexes operate as classical catalysts driving GluA-CNIH/TARP assembly that can also occur in their absence, albeit with less efficiency. As a consequence of this mechanism, the number of AMPARs at the plasma membrane can be effectively regulated as seen in knockdown and overexpression experiments (Figs 6 and 7). It is noteworthy that the presented model does not preclude any additional factors described for ER exit of GluA and/or GluA-CNIH/TARP assemblies[40,41] nor does it rule out additional regulatory processes occurring in the secretory pathway or along vesicle exocytosis/endocytosis. Importantly though, the latter cannot compensate for the disruption of the ER-priming complex (as induced by sh-RNA-mediated knockdown of FRRS1l or CPT1c or by the disease mutations in FRRS1l, Figs 6–8).

Our experimental results have some additional noteworthy implications. First, they emphasize the potential significance of functionally uncharacterized, including purely intracellular, constituents of the AMPAR proteome for assembly and function of this key component of excitatory synaptic transmission. Second, they demonstrate that the proteome constituents do not assemble randomly, but rather in 'groups' as shown here for

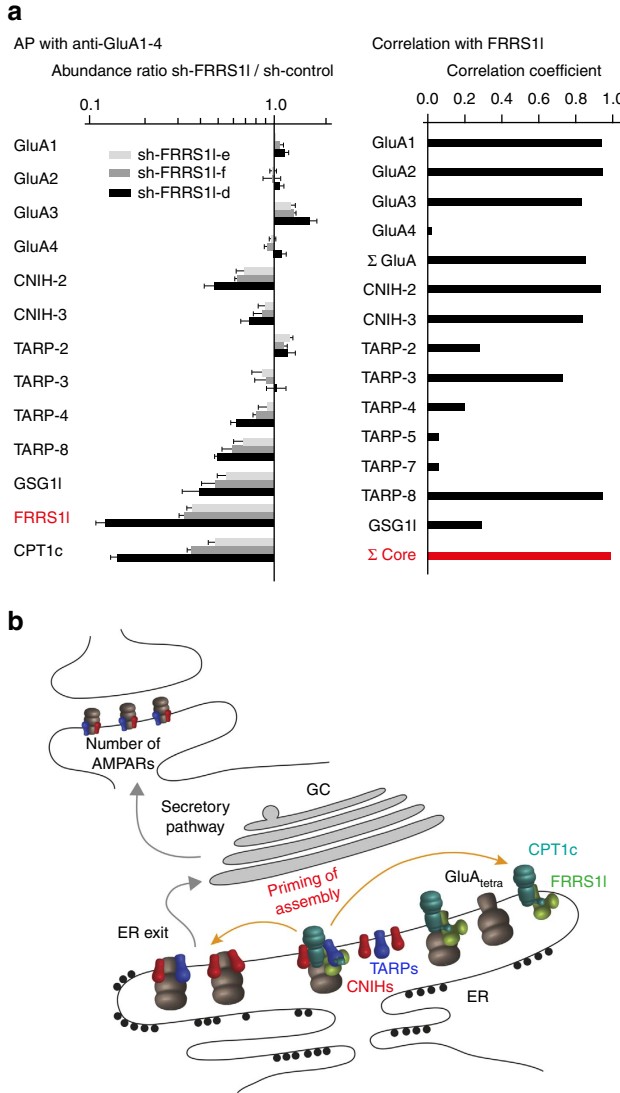

**Figure 8 | Operation of FRRS1l/CPT1c complexes in AMPAR biogenesis in the ER.** (**a**) Left panel: Abundance ratios (mean ± s.e.m. of four measurements) as in Fig. 2b determined in depleting GluA1-4 APs from neuronal cultures transduced with three additional sh-RNAs directed against distinct sequence stretches on FRRS1l or with sh-control. Note the different efficiencies of the three sh-FRRS1ls on both FRRS1l and the majority of core subunits and CPT1c. Right panel: correlation analysis of FRRS1l with GluA1-4 and the core subunits of AMPARs across AP data sets from different brain regions (data from ref. 15 re-evaluated). Note that maximal correlation is observed between FRRS1l and the sum of all core subunits (red bar) rather than with the sum of GluAs or individual subunits. (**b**) Scheme summarizing the proposed role of FRRS1l as a 'catalyst' in AMPAR biogenesis and delivery of synaptic AMPARs. Newly synthesized GluA tetramers (GluA$_{tetra}$) co-assemble with FRRS1l–CPT1c complexes during early biogenesis in the ER. These GluA-FRRS1l/CPT1c complexes prepare subsequent co-assembly with the inner core subunits TARPs and CNIHs depicted as a short-lived intermediate priming complex ('priming of assembly'). After binding of CNIHs/TARPs, FRRS1l–CPT1c complexes dissociate (arrow to the right), and the resulting GluA-CNIH/TARP assemblies represent AMPARs competent for ER exit (arrow to the left) and delivery to the plasma membrane. Stoichiometries were not implicated in neither of the illustrated protein assemblies.

FRRS1l–CPT1c and their partner proteins (Figs 1 and 2) and as suggested by the clustered correlation analysis of the AMPAR constituents[15].

The significance of FRRS1l–CPT1c-containing AMPARs in the context of the whole brain is reflected by both the phenotype of CPT1c knockout mice presenting with impaired spatial learning, reduced muscular strength and hypoactivity[23,42], but in particular by the severe clinical features observed in patients carrying homozygous mutations in the *FRRS1L* gene (Fig. 3; refs 27,28). All of these patients exhibited markedly impaired brain function with severe intellectual disability and epilepsy.

A few of these features may be reconciled by the properties of FRRS1l expression and the results of its knockdown in adult rats (mimicking the lack of FRRS1l-GluA assembly; Figs 6 and 7). Thus a decrease in EPSC amplitudes in most (if not all) excitatory glutamatergic synapses is expected to impact both synaptogenesis and signal transmission and, consequently, the processes underlying synaptic plasticity involved in skill learning and memory formation. Furthermore, the reduced EPSCs should, via imbalance between excitatory and inhibitory synaptic transmission, promote the onset of seizures[43,44]. And finally, the regression phenotype observed in patients of families B and C might correlate with the pronounced upregulation observed for FRRS1l expression during postnatal development[15], while the milder phenotype segregating with the K155E mutation (family A) was accompanied by a less impaired interaction of FRRS1l with GluA$_{tetra}$ (Fig. 4d).

In conclusion, our work provides *de novo* assignment of cellular function(s) to FRRS1l–CPT1c complexes, previously uncharacterized constituents of the AMPAR proteome, that impact receptor biogenesis and fast excitatory synaptic transmission. Our work emphasizes not only the importance of the AMPAR proteome but also necessitates further extensive research to understand its individual components in terms of overall AMPAR biology and its significance for encoding the molecular framework underlying the operation and dynamics of excitatory neurotransmission in the brain.

## Methods
**Molecular biology.** The cDNAs used were all verified by sequencing and had the following GenBank (www.ncbi.nlm.nih.gov/genbank) accession numbers: M38060.1 (GluA1i, flip variant of GluA1), NM_017261.2 (GluA2i), NM_053351 (TARP-2), NM_001025132 (CNIH-2), AF357970.1 (CPT1c), NM_014334 (FRRS1l), and BC117752.1 (Sac1).

**Biochemistry and cell biology.** *Affinity purification from brain membranes.* Plasma-membrane-enriched protein fractions were prepared from freshly isolated WT rat or mouse brains (pools of 10–20 animals) or from three mouse brains of CPT1c knockout animals[23], as previously described[16], and solubilized with buffers CL-47 and CL-91 (Logopharm GmbH, Germany) for 30 min on ice (at 1 mg protein per ml). After clearing by ultracentrifugation (10 min, 150,000g), solubilisates were incubated with the respective immobilized ABs. After 2 h of incubation and two brief washes, proteins were eluted, shortly run on SDS–PAGE gels and silver-stained. Lanes were cut into two sections (high and low MW) and digested with sequencing-grade modified trypsin (Promega, Mannheim, Germany). Peptides were extracted and prepared for MS analysis as described[15,16]. The following ABs were used: anti-GluA1 (Millipore, #AB1504, 1:1,000), anti-GluA2 (NeuroMab, #75-002, 1:1,000), anti-GluA2/3 (Millipore, #07-598, 1:1,000), anti-GluA3 (Synaptic Systems, #182203, 1:1,000), anti-GluA4 (Millipore, #AB1508, 1:1,000), rabbit anti-FRRS1l-a, anti-FRRS1l-b (epitope: rat FRRS1l aa 47-66, AB generation by AbFrontiers, South Korea, 1:1,000, 15 µg), rabbit anti-FRRS1l-c, anti-FRRS1l-d (rat FRRS1l aa 27–44, generated by AbFrontiers, 1:1,000, 15 µg), anti-CPT1c-a (Santa Cruz Biotechnology, sc-139479, 1:1,000, 15 µg), anti-CPT1c-b[45], anti-TARP-a (anti-TARP-γ2,3; Millipore, #07-577, 15 µg), anti-TARP-b (anti-TARP-γ 2,4,8; NeuroMab, #75-252, 15 µg), anti-TARP-c (anti-TARP-γ8; Frontier Institute, #TARP γ8-Rb-Af1000, 15 µg), and IgG rabbit (Millipore, #12-370, 15 µg). Two anti-FRRS1l and anti-CPT1c APs used mixtures of ABs (anti-FRRS1l-a,b,c; anti-CPT1c-a, b).

*Target-depleted solubilisates as source for anti-FRRS1l-c APs.* Rat membrane proteins (1.5 ml) solubilized with CL-47 and CL-91 (1 mg ml$^{-1}$) were incubated for 2 h with 120 µg of a mixture of immobilized anti-FRRS1l-a and anti-FRRS1l-b. Thereafter, ABs were removed and the supernatant, deprived of at least 95% of FRRS1l protein, was subsequently incubated for 2 h with anti-FRRS1l-c ABs. After brief washing, proteins were eluted with SDS buffer without dithiothreitol. These

eluates served as a negative control (specificity determination) in APs with the respective AB (see 'Protein quantification').

*Two-step APs.* Rat and mouse brain solubilisates (0.5 ml, concentration 1 mg ml$^{-1}$) were incubated for 2 h with a mixture of immobilized FRRS1l-targeting ABs (anti-FRRS1l-a, -b, -c; 80 µg). Subsequently, the supernatants were subjected to APs with a mixture of anti-GluA1-4 ABs (see above, 80 µg); bound proteins were processed as above. These experiments were also performed in reverse order: An AP using the anti-GluA1-4 AB mix (80 µg) followed by an AP with the mixture of anti-FRRS1l ABs (80 µg). Efficiency of the APs was verified by western blot analysis of SDS–PAGE-resolved samples taken before and after each incubation step (Supplementary Fig. 1). The eluates were processed for MS analysis as described above. APs shown in Fig. 1 were performed 2–4 times.

*AMPARs in neuronal cultures after FRRS1l knockdown.* Primary cortical neurons were prepared from rats at E18 and cultured essentially as described[46]. Cells were transduced with lentivirus at DIV1 and cultivated for 3 weeks. Equal amounts of neurons were treated with sh-control, sh-FRRS1l-d, sh-FRRS1l-e or sh-FRRS1l-f (see 'Generation of lentivirus'). The rate of infection was estimated based on the fluorescence of the enhanced green fluorescent protein (EGFP) marker included into the viral vectors. Neurons were harvested and crude membranes were prepared as described[35]. Membranes were solubilized with CL-47 and incubated with a mixture of immobilized anti-GluA1-4 ABs. The eluates were subsequently processed for MS analysis as described above.

*AMPARs in CPT1c knockout versus WT mice.* Mouse brain solubilisates (0.5 ml, solubilized with CL-47, concentration 1 mg ml$^{-1}$) from WT and CPT1c knockout animals were incubated for 2 h with a mixture of immobilized anti-GluA1-4 ABs (see above, 50 µg). Efficiency of the APs was verified by western blot analysis of SDS–PAGE-resolved samples taken before and after incubation. The eluates were processed for MS analysis as described above.

*Reconstitution of protein complexes.* The indicated proteins were expressed in transiently transfected tsA-201 cells that had been cultured according to manufacturer's advice in DMEM high glucose medium supplemented with 10% FCS, 1% HEPES 1 M Buffer and 1% Penicillin/Streptomycin 10,000 U ml$^{-1}$. Two days after transfection (polyethylenimine transfection, Polysciences, USA), cells were washed and lysed in homogenization buffer (320 mM sucrose, 10 mM Tris/HCl pH 7.5, 1.5 mM MgCl$_2$, supplemented with protease inhibitors) by sonication. After clearing by centrifugation (1,000g for 10 min), membrane vesicles were pelleted by ultracentrifugation (125,000g, 20 min) and solubilized with CL-47. Solubilized protein complexes were affinity-purified with anti-GluA1 or anti-FRRS1l-a ABs as described above. Aliquots of solubilisates (input) and eluates were resolved by SDS–PAGE, blotted onto PVDF-membranes and decorated with target-specific ABs: anti-GluA1 (Millipore, 1:1,000), anti-FRRS1l-a (1:1,000), and anti-CPT1c (Santa Cruz Biotechnology, 1:1,000). Anti-HA (Roche, #11867423001, 1:1,000) was used for targeting HA-tagged SAC1. Horseradish peroxidase-conjugated secondary ABs in combination with ECL Prime (GE Healthcare, Germany) were used for visualization. Images of respective western blots (Fig. 4) have been cropped for presentation. Full size images are presented in Supplementary Figs 10 and 11. All APs were done at least three times.

*BN–PAGE analysis.* Two-dimensional BN–PAGE/SDS–PAGE separations were essentially done as described[16]. Protein complexes from the rat brain were solubilized in CL-47 and insoluble proteins were removed by centrifugation (125,000g, 10 min). Solubilisates were thereafter centrifuged (400,000g, 60 min) on a sucrose gradient (50% sucrose cushion) to replace salt by 0.5 M betaine. Samples were separated on a linear 3%–15% polyacrylamide gradient gel in 15 mM BisTris/50 mM Tricine/0.01% Coomassie G250 running buffer and 15 mM BisTris (pH 7.0) as anode buffer. Rat mitochondrial membrane protein complexes were use as a standard for complex size in the first dimension. Excised BN–PAGE lanes were incubated for 15 min in Laemmli buffer and placed on top of 10% SDS–PAGE gels. After electroblotting on polyvinylidene difluoride membranes, the blot was cut horizontally into different MW ranges and stained with the indicated ABs.

*Surface biotinylation assay.* Transiently transfected tsA-201 cells were briefly washed with PBS and surface-labelled by adding 10 mM EZ-Link Sulfo-NHS-SS-Biotin (Thermo Scientific) dissolved in PBS and incubated for 20 min at room temperature. After quenching the reaction by adding 50 mM Tris/HCl buffer and three washing steps, cells were harvested and lysed and membrane vesicles were prepared as described above. Solubilisates were obtained with CL-91 and incubated with Pierce NeutrAvidin Agarose (Thermo Scientific) for 2 h. Beads were washed three times with CL-91 and bound proteins were eluted with Laemmli buffer. Equivalent amounts of solubilisates (total, To in Supplementary Fig. 4a) and supernatants after NeutrAvidin treatment containing all non-biotinylated (intracellular, IN) proteins together with the biotinylated surface fraction (S, 12.5-fold enriched) were western blot analysed as above (using anti-Calnexin (Abcam, #75801) for additional detection). All surface biotinylation assays were performed 3–5 times.

*Analysis of FRRS1l primary sequence.* tsA-201 cells were transiently transfected with CPT1c and FRRS1l WT or FRRS1l C-terminally tagged with a HA-His6 fusion (FRRS1l$_{C-tag}$). After 2 days in culture, cells were harvested and membranes were prepared as described above. Proteins were solubilized with CL-91 and affinity-purified with anti-FRRS1l-a. Bound proteins were eluted, separated on SDS–PAGE gels and silver-stained. Narrow protein bands containing FRRS1l were excised, bisected and digested either with sequencing-grade modified trypsin (#V5113, Promega, USA) or with α-lytic proteases (WaLP, Sigma-Aldrich #A6362

and MaLP (M190A) Sigma-Aldrich #A6487, ref. 47). Samples were further processed for MS analysis as described below. In parallel, aliquots of membrane fractions and AP eluates were processed for control western blot analysis; FRRS1l protein isoforms were visualized by an anti-His AB (AbD Serotec #MCA1396, 1:1,000) and subsequently with anti-FRRS1l-a.

**Mass spectrometry.** *LC-MS/MS analysis.* Mass spectrometric analysis was carried out as described in ref. 15. Peptide samples dissolved in 0.5% trifluoroacetic acid were loaded onto a trap column (C18 PepMap100, 5 µm particles; Thermo Scientific) with 0.05% trifluoroacetic acid (20 µl min$^{-1}$ for 5 min), separated by reversed phase chromatography via a 10 cm C18 column (PicoTip Emitter, 75 µm, tip: 8 µm, New Objective, self-packed with ReproSil-Pur 120 ODS-3, 3 µm, Dr Maisch HPLC; flow rate 300 nl min$^{-1}$) using an UltiMate 3000 RSLCnano HPLC system (Thermo Scientific) and eluted by an aqueous organic gradient (eluent 'A': 0.5% acetic acid; eluent 'B' 0.5% acetic acid in 80% acetonitrile; 'A'/'B' gradient: 5 min 3% B, 60 min from 3% B to 30% B, 15 min from 30% B to 99% B, 5 min 99% B, 5 min from 99% B to 3% B, 15 min 3% B). Sensitive and high-resolution MS analyses were done on an Orbitrap Elite mass spectrometer with a Nanospray Flex Ion Source (both Thermo Scientific). Precursor signals (LC-MS) were acquired with a target value of 1,000,000 and a nominal resolution of 240,000 (full-width at half-maximum) at m/z 400; scan range 370–1,700 m/z. Up to 10 data-dependent CID fragment ion spectra (isolation width 1.0 m/z with wideband activation) per scan cycle were allowed in the ion trap with a target value of 10,000 (maximum injection time 200 ms for complex mixtures and 400 ms for gel bands) with dynamic exclusion (exclusion duration 30 s; exclusion mass width ± 20 p.p.m.), preview mode for FTMS master scans, charge state screening, monoisotopic precursor selection and charge state rejection (unassigned charge states and for trypsin-digested samples also charge state 1) enabled. For highly reliable peptide identification in excised FRRS1L-containing gel bands, additional high-resolution CID fragment ion spectra of α-lytic protease-digested samples were acquired in the Orbitrap analyser with a target value of 50 000 (maximum injection time 500 ms) and a nominal resolution of 15,000 (full-width at half-maximum at m/z 400; centroid data). The mass spectrometric data have been deposited to the ProteomeXchange Consortium via the PRIDE partner repository with the data set identifier PXD006413 and 10.6019/PXD006413.

*Protein identification.* LC-MS/MS data were extracted using 'msconvert.exe' (part of ProteoWizard; http://proteowizard.sourceforge.net/, version 3.0.6906). Peak lists were searched against a modified UniProtKB/Swiss-Prot database (release 2017_01 for Figs 1b, 2b and 8a, release 2015_09 or newer for remaining analyses; all rat, mouse and human entries, as well as sp|P02769, sp|P00766 and sp|P00761, supplemented with the TrEMBL/NCBI entries tr|D3ZVQ3, tr|D4A4M0, tr|M0RB53, tr|D4A0X1 and XP_008765687.1 for missing AMPAR proteome constituents) using Mascot 2.6.0 (Matrix Science, UK). Initially, preliminary searches with high peptide mass tolerance ( ± 50 p.p.m.) were performed. After linear shift mass recalibration using in-house developed software, peptide mass tolerance was reduced to ± 5 p.p.m. for final searches. Fragment mass tolerance was set to ± 0.8 Da (ion trap MS/MS spectra). One missed trypsin cleavage and common variable modifications including S/T/Y phosphorylation were accepted for peptide identification. Significance threshold was set to $P < 0.05$. Proteins identified by only one specific MS/MS spectrum or representing exogenous contaminations such as keratins or immunoglobulins were eliminated.

*Primary sequence analysis.* Peak lists extracted from the LC-MS/MS data of the FRRS1l protein band were finally searched against databases containing all human UniProtKB/Swiss-Prot entries, the heterologous FRRS1l and CPT1c sequences (see above) and a series of C-terminally truncated FRRS1l sequences (truncation after P316, S317, A318, A319, Y320, Q321, T322, F323, S324 or S325 to promote Mascot identification of the C-terminal peptide of the truncated protein in the lower MW band). For high-resolution MS/MS spectra acquired in the Orbitrap analyser, the fragment mass tolerance was reduced to ± 20 m.m.u. One missed trypsin cleavage and up to five missed WaLP/MaLP cleavages (after A, S, T or V/A, F, L, M, T or V, respectively) were allowed for the respective digests. Singular peptide matches with poor Mascot identification scores or expect values were manually checked before the respective peptide sequences were integrated in the coverage shown in Supplementary Fig. 6.

*Protein quantification.* Label-free quantification of proteins was based on peak volumes (PVs = peptide m/z signal intensities integrated over time) of peptide features as described previously[30]. Peptide feature extraction was done with MaxQuant (http://www.maxquant.org/[48], version 1.4) with integrated effective mass calibration. Features were then aligned between different LC-MS/MS runs and assigned to peptides with retention time tolerance ± 1 min and mass tolerance: ± 1.5 p.p.m. using an in-house developed software. The resulting peptide PV tables formed the basis for protein quantification (molecular and relative abundance in Figs 1, 2 and 8 and Supplementary Figs 1b and 5c).

Molecular abundances of AMPAR constituents (Fig. 1a) were determined from protein profiles calibrated with a label-free QconCAT method[15,16]. First, the peptide PV data for each protein were filtered for outliers and false-positive assignments using a recently developed correlation-based method[30]. For each peptide, the PVs were then normalized to their maximum over all AP data sets yielding relative peptide profiles, ranked for each protein by pairwise Pearson correlation. The medians of at least 2–7 or, for larger proteins, the 50% best

correlating protein-specific peptide profiles were used to calculate the relative abundance values of each protein profile as described in ('Protein profiles'[30]). These protein profiles were scaled to best fit the peptide calibration values obtained from MS analyses of fusion protein standards[15,16] to obtain molecular abundance values (arbitrary units) for each protein. These values were finally normalized to the molecular abundance of the primary target protein(s) in each AP data set to obtain the degree of association with the target. Protein abundances in Fig. 2a (which included proteins for which QconCAT standards were not available) were estimated from abundance$_{norm}$(spec) values determined as the sum of all assigned and protein isoform-specific PVs divided by the number of MS-accessible protein isoform-specific amino acids[49]).

For relative quantification of proteins in two different groups of samples (replicate measurements in Figs 1b, 2b and 8a and Supplementary Fig. 1b), protein ratios (rPVs) were calculated after normalization of each AP data set to the (relative) abundance of its primary target (that is, molecular abundance of total AMPARs (Figs 1b and 2b, Supplementary Fig. 1b) or relative abundance of all AMPAR subunits (Fig. 8a) as follows: For the two-step APs (Fig. 1b and Supplementary Fig. 1b), rPVs were calculated for each first replicate AP versus the averaged PVs of the second replicate APs using the TopCorr method[49]. The relative amount ($F$) of the protein found in the first AP was determined by dividing each rPV value by the sum of (rPV + 1) and given as mean ± s.e.m. The fraction of the protein in the second AP was defined as $(1 − F)$. Protein ratios in Figs 2b and 8a were determined from the protein profiles (determined as described above); for Fig. 2b, reference was the mean of four APs from WT, and for Fig. 8b, reference was the mean of four APs from control sh-RNA-transfected cultures. A two-sided Student's $t$-test was performed to determine significance of the observed changes. For the APs in Fig. 2a, specificity threshold ratios were determined from rPV histograms of all proteins detected in the respective AP versus control[49]. Proteins were considered specifically co-purified with FRRS1l when (i) rPV(WT rat versus IgG)/threshold(versus IgG) was >1 in experiments with two independent ABs and (ii) the control purification using a target protein-depleted solubilisate (obtained for anti-FRRS1l-c) showed rPV(WT rat versus the biochemical depletion control)/threshold(versus depletion control) >1. Similarly, specificity of co-purification with CPT1c was determined based on criteria (i) rPV ratio (WT rat versus IgG)/threshold(versus IgG) >1 for all experiments (two independent CPT1c ABs) done from WT rat and WT mouse compared to IgG, and (ii) rPV ratio (WT mouse versus the CPT1c knockout)/threshold(versus knockout control) >1 for both ABs.

**Genetic analyses.** Institutional research ethical approval and written consent were obtained for all participants in the study (Comité de Protection des Personnes (CPP) 'Ile-de-France II', Hospital Necker-Enfants malade, 45 Rue des Saint-Pères, 75006 Paris and Ethics Committees of the University of Bonn (Ethikkommission, Reuterstr. 2B, 53113 Bonn, Germany). For family A, exome capture was performed at the genomic platform of the foundation IMAGINE (Paris, France) with the SureSelect Human All Exon Kit (Agilent Technologies) on all affected individuals (II.1, II.4., II.5) and their healthy parents. Single-end sequencing was performed on an HQ Illumina Genome Analyzer IIx (Illumina) generating 72 base reads. Sequence data were analysed by the Bioinformatic platform and visualized via the interface created by the Bioinformatic platform (Université Paris Descartes, Paris). For sequence alignment, variant calling and annotation, the sequences were aligned to the human genome reference sequence (UCSC Genome Browser, GRCh38 build) by BWA aligner[50]. Downstream processing was carried out with the Genome Analysis Toolkit (GATK)[51], SAMtools[52] and Picard Tools. Substitution calls were made with GATK Unified Genotyper, whereas indel calls were made with a GATK IndelGenotyperV2. All calls with a read coverage ≤2× and a Phred-scaled single-nucleotide polymorphism (SNP) quality of ≤20 were filtered out. All the variants were annotated with an in-house-developed annotation software. A single shared homozygous variant emerged and filtering for compound heterozygous variants, X-linked or shared heterozygous *de novo* variants did not provide any additional candidate.

For family B, SNP microarrays (Affymetrix SNP Chip 6.0, Santa Clara, USA) were run for all affected individuals and healthy siblings and positional mapping was performed as described previously[53]. Three regions identical by descent with a total length of 27 Mb were identified. Exome sequencing was performed using DNA from individual II-8. DNA was enriched using the SureSelect Human All Exon 50 Mb Kit (Agilent technologies) and paired-end sequenced on a SOLiD 4 instrument (Life Technologies, USA). Details on sequencing procedure and bioinformatics processing are described in ref. 54. Filtering of candidate variants consisted of three steps: (i) common (minor allele frequency >1%) genetic variants (by public databases (ExAc, EVS, 1000GP) and in-house databases) were excluded, (ii) homozygous variants in the candidate regions were selected, and (iii) checked for quality (manual inspection of the aligned reads and Sanger sequencing where needed), pathogenicity (based on minor allele frequencies in different populations on *in silico* predictions and conservation) and causality (gene functions and disease associations). PCR and Sanger sequencing were used to exclude technical artifacts and validate segregation.

For family C, exome sequencing was performed using DNA samples of the affected child and his healthy parents (during exome sequencing at the Centogene laboratories, Rostock, Germany). Genomic DNA was amplified with the Ion

AmpliSeq Exome Kit. The DNA Libraries were pooled, barcoded and sequenced using an Ion Torrent Proton sequencer. Variant calling and filtering strategies were as described[55]. Briefly: (i) common genetic variants based on available public databases (ExAc, EVS, 1000GP) were excluded, (ii) remaining variants were evaluated for all inheritance patterns (autosomal-dominant due to a *de novo* mutation, autosomal-recessive due to homozygous or compound heterozygous mutations and X chromosomal recessive inheritance), and (iii) variants segregating with the symptoms and predicted to impact the protein structure (missense, splicing, insertions or deletions and so on) were selected. From this pool, variants observed in >2% in our in-house database (>4000 exomes) were excluded and the remaining candidate variants were checked for quality, pathogenicity and causality as mentioned above. Conventional Sanger sequencing confirmed candidate variants; segregation of these variants with the disease was assessed for all available family members.

**Immunohistochemistry.** After fixation with paraformaldehyde, transverse hippocampal slices (60 µm thick) were obtained from the brains of 3–4-week-old Wistar rats. Brain slices and tsA-201 cells (culturing and transfection as above) were blocked with 6% normal goat serum in 0.1 M phosphate buffer. Proteins were immunodetected after permeabilization with 0.1% Triton-X100, by incubation with the following target-specific primary ABs: anti-FRRS1l-a, anti-FRRS1l-c, anti-vGlut1 (NeuroMab, #73-066), anti-CPT1c (sc-139480, Santa Cruz Biotechnology), anti-CPT1c (sc-393070, Santa Cruz Biotechnology), and anti-Calnexin (ab140818, Abcam). Mitochondria were detected with the mitotracker Deep Red FM (Molecular Probes, #M22426). Corresponding secondary ABs conjugated to Alexa 488, 555 or 633 (Molecular Probes, USA) were incubated for 1 h. Brain slices and coverslips were mounted in Fluor Save reagent (Calbiochem). All experiments in Fig. 4 were done three times as independent transfections; immunostainings in slices (Fig. 5) were performed with two different animals. For verification of protein knockdown by sh-RNAs, neurons were filled during whole-cell recording with 0.1% biocytin (Molecular Probes, USA) added to the intracellular solution. After recordings, slices were fixed overnight at 4 °C in 0.1 M phosphate buffer containing 4% paraformaldehyde[15]. Immunofluorescence was analysed using a confocal laser-scanning microscope (LSM 710 meta, Zeiss). Confocal images were acquired with a Plan-Apochromat 40 × /1.3 N.A. and 63 × /1.4 N.A. oil objectives (Zeiss). Line scans were done with the Zen 2012 SP1 software (Zeiss, Germany).

**Electron microscopy.** For pre-embedding immunogold labeling of FRRS1l, perfused tissues from two Wistar rats were prepared as described previously[56]. Sections (50 µm) from the CA1 area of the hippocampus were cryo-protected and freeze-thawed, then incubated in 20% normal goat serum (NGS; Vector Laboratories) and incubated with anti-FRRS1l-a (2.6 µg ml$^{-1}$) diluted in Tris-buffered saline (TBS) containing 3% NGS for 24 h at 4 °C. Subsequently, sections were incubated with goat anti-rabbit secondary AB (Fab fragment, diluted 1:100) coupled to 1.4 nm gold (Nanoprobes, Stony Brook, NY), made up in TBS containing 1% NGS, overnight at 4 °C. After washes in TBS, sections were washed in double-distilled water followed by silver enhancement of gold particles with an HQ Silver Kit (Nanoprobes, USA) for 10 min. After several washes in phosphate buffer (PB) sections were treated with 1% OsO$_4$ in PB for 40 min, washed in PB and double-distilled water and then contrasted in 1% uranyl acetate for 40 min. Subsequently, they were dehydrated in a series of ethanol and propylene oxide and flat embedded in epoxy resin (Durcupan ACM; Fluka, Switzerland). After polymerization, ultrathin sections (70 nm) were cut using an ultramicrotome (Reichert Ultracut E; Leica, Vienna, Austria) and analysed in an electron microscope (Philips CM100).

**Electrophysiology.** *In vivo stereotactic injection.* The distinct lentiviruses were injected into Wistar rats 6-7 days after birth (P6–P7). Animals were anaesthetized by injection of a ketamine/dorbene mixture and mounted in a Kopf stereotaxic frame (Tujunga, USA). Virus-containing solution (0.5–2 µl) was injected at a single site targeting the hippocampus by means of a UMP3 controller (WPI, Sarasota, USA) and a nanofil syringe/needle (WPI, Sarasota, USA). Following surgery, pups recovered rapidly by antagonist injection and were returned to their home cage. Recordings were performed 10–18 days following virus injection. Animal proce-dures were in accordance with national and institutional guidelines and approved by the Animal Care Committee Freiburg according to the Tierschutzgesetz (AZ G-12/47).

*Generation of lentivirus.* The lentivirus driving expression of sh-RNAs or FRRS1l protein were generated as detailed in ref. 15. Briefly, oligonucleotides targeting rat FRRS1l (5′-TTGTGGATCTGCACTTGAG-3′, sh-FRRS1l; 5′-GCCAGGCTGTAATGCAGAA-3′, sh-FRRS1l-b; 5′-TTGTGGATCTGCACT TGAG-3′ and 5′-GCCAGGCTGTAATGCAGAA-3′, sh-FRRS1l-c; 5′-AGGTCAA GCTTTAGTGAAA-3′, sh-FRRS1l-d; 5′-GGATTTAAACCCAGACAAA-3′ sh-FRRS1l-e; 5′-GCACAACAGTCGATCTATA-3′, sh-FRRS1l-f), rat CPT1c (5′-GCTGGCATTGGTCAGAATC-3′, sh-CPT1c; 5′-GGAGGAGTTCAGGTTGG AAA-3′, sh-CPT1c-b), rat TARP-8 (5′-CGGAGGACACGGACTACA-3′ and 5′-CCGTCAACATCTACATCGA-3′; sh-TARP-8), rat TARP-2 (5′-CCGAAG ACGCGGACTACGA-3′; sh-TARP-2) and the control oligonucleotide (5′-TCGC TTGGGCGAGAGTAAG-3′; sh-control) that does not target any gene transcript[57]

were synthesized as sense–antisense hairpins, subcloned into pSuper (OligoEngine) and then transferred to viral vectors (FUGW[58]) equipped with enhanced GFP. For exogenous (over)expression, FRRS1l was subcloned into a double promoter lentivector (System Biosciences, #CD511B-1) with copGFP as expression marker. Lentiviruses were generated by transfecting tsA-201 cells with transfer (pFUGW) and packaging (pVSV and pΔ8.9) vectors. The medium was collected after 72 h and filtered and virus particles were suspended in artificial cerebrospinal fluid providing stock solutions with a titer of $10^7$ $10^8$ ml$^{-1}$.

*Slice preparation.* Transverse 300-μm-thick hippocampal slices were cut from the brains of 3–4-week-old Wistar rats, as described[59]. Hippocampal slices were cut in ice-cold sucrose-containing physiological saline using a commercial vibratome (VT1200S, Leica Microsystems). Slices were incubated at 35 °C, transferred to a recording chamber and superfused with physiological saline at room temperature. Cells and subcellular compartments (MFBs) were visualized by infrared differential interference contrast (IR-DIC) video-microscopy using an Axio examiner microscope (Zeiss, Germany) equipped with a 63 × water-immersion objective coupled to an epi-fluorescence system.

*Cellular and subcellular patch-clamp recording.* Patch pipettes were pulled from borosilicate glass (Hilgenberg, Germany; outer diameter, 2 mm; wall thickness, 0.7 mm for presynaptic recordings and 0.5 mm for somatic recordings). When filled with internal solution, they had resistances of ∼15 MΩ (presynaptic pipettes) and 4–8 MΩ (postsynaptic pipettes). Patch pipettes were positioned using two Kleindiek micromanipulators (Kleindiek Nanotechnik, Germany). A Multiclamp 700B amplifier (Molecular Devices, USA) was used for recordings. Pipette capacitance of both electrodes was compensated to 70–90%. Voltage and current signals were filtered at 10 kHz with the built-in low-pass Bessel filter and digitized at 20 kHz using a Digidata 1440A (Molecular Devices, USA). The pClamp10 software (Molecular Devices, USA) was used for stimulation and data acquisition.

For dissection and storage of slices, a sucrose-containing physiological saline containing 87 mM NaCl, 25 mM NaHCO$_3$, 25 mM D-glucose, 75 mM sucrose, 2.5 mM KCl, 1.25 mM NaH$_2$PO$_4$, 0.5 mM CaCl$_2$ and 7 mM MgCl$_2$ was used. Slices were superfused with physiological extracellular solution that contained 125 mM NaCl, 25 mM NaHCO$_3$, 2.5 mM KCl, 1.25 mM NaH$_2$PO$_4$, 1 mM MgCl$_2$, 2 mM CaCl$_2$ and 25 mM glucose (equilibrated with a 95% O$_2$/5% CO$_2$ gas mixture). Pipettes were filled with a K-methylsulfonate intracellular solution containing 120 mM KMeHSO$_3$, 20 mM KCl, 2 mM MgCl$_2$, 2 mM Na$_2$ATP, 10 mM HEPES and 0.1 mM EGTA. Membrane potentials are given without correction for liquid junction potentials. Values given throughout the manuscript indicate mean ± s.e.m. Significance of differences was assessed by a nonparametric Mann–Whitney U-test.

*Single and paired recordings.* Simultaneous recordings were established between the soma of a MC and one MFB located in apposition to its dendrites (with a maximal distance of 60 μm from the soma). Both bouton-attached and whole-bouton configurations were used for eliciting action potentials that evoked synaptic transmission; action potentials were elicited by a brief current pulse (2 ms, 200 pA). The postsynaptic neuron was held at − 70 mV, by injecting 0 to − 200 pA holding current. EPSCs in individual MFB–MC or MFB–interneuron synapses were determined as averages of 3–30 evoked EPSCs; thereby, latency of evoked EPSCs (delay between presynaptic AP and EPSC onset defined as 5% of peak amplitude) was between 0.2 and 2 ms. Spontaneous EPSCs were recorded from postsynaptic neurons that were held at − 70 mV. Ratios of AMPA-to-NMDA currents were determined from EPSCs elicited by spontaneous APs and recorded at a membrane potential of 40 mV; AMPAR-mediated currents were blocked by 20 μM CNQX to obtain the NMDAR-mediated EPSC component that was finally verified through its complete block by 50 μM D-APV. As a control, AMPAR-mediated EPSCs were recorded in the same cells at a holding potential of − 70 mV. EPSCs and currents through extra-synaptic (somatic) AMPARs (Fig. 7c) were determined with the following procedure: After recording spontaneous EPSCs in whole-cell configuration (− 70 mV), the patch-pipette initially placed on the soma of the MC was excised into out-outside configuration and placed in front of a piezo-driven fast application system that enables solution exchanges within <100 μs (20–80%, measured from the open tip response during a switch between normal and 10 × diluted extracellular solution). AMPAR currents were elicited by 100-ms or 1-ms applications of 10 mM glutamate.

Data for all conditions (non-infected and virally transduced) were recorded in brain slices obtained from at least three different animals from three different pups; recordings from virally transduced and non-infected control cells were performed in the same slices.

*Data analysis.* Stimfit 0.9 software and Igor Pro (WaveMetrics, USA) were used to analyse data from spontaneous and evoked EPSCs. The rise time $_{20–80\%}$ was determined as the time interval between the points corresponding to 20% and 80% of the peak amplitude. The peak current was determined as the maximum within a 2-ms window following the presynaptic action potential. For both single and paired recordings, the EPSC decay time constant ($\tau_{EPSC\ decay}$) was obtained from a mono-exponential function fitted to the decay phase of the current[15]. The analysis was restricted to spontaneous and evoked EPSCs fulfilling the following criteria: (1) amplitude was >20 pA (more than twofold larger than the noise recording), (2) decay was complete, that is, traces declined to baseline and were not interfered by further synaptic events (that is, ≥40 ms), (3) rise time $_{20–80\%}$ was between 0.4 and 2.0 ms.

*Patch recordings from culture cells.* Recordings from outside-out patches excised from Chinese hamster ovary cells were performed at room temperature (22–24 °C). Currents were recorded with an EPC10 amplifier (HEKA, Germany), low-pass

filtered at 3 kHz and sampled at 5–20 kHz. Recording pipettes made from quartz glass had resistances of 1–2 MOhm when filled with (in mM): 135 CsF, 33 CsOH, 2 MgCl$_2$, 1 CaCl$_2$, 0.1 spermine, and 11 EGTA, pH 7.4. The extracellular solution applied to outside-out patches contained (in mM): 144 NaCl, 5.8 KCl, 0.9 MgCl$_2$, 1.3 CaCl$_2$, 0.1 NaH$_2$PO$_4$, 5.6 D-glucose, and 10 HEPES (pH 7.4). Rapid application/removal of glutamate (10 mM, dissolved in extracellular solution) was performed using a Piezo-controlled fast application system with a double-barrel application pipette that enables solution exchanges within <100 μs (20%–80%, measured from the open tip response during a switch between normal and 10 × diluted extracellular solution). Deactivation, desensitization and recovery from desensitization of AMPARs were characterized by time constants derived from mono-exponential fits to the decay phase or recovery of the glutamate-activated currents. Curve fitting and data analysis were done with Igor Pro (MaveMetrics, USA). Data in text and figures are given as mean ± s.e.m.

**Data availability.** The mass spectrometry proteomics data generated during this study are available via ProteomeXchange with identifier PXD006413; any other data supporting the findings of this study are available from the corresponding authors upon reasonable request.

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

## Acknowledgements

We are grateful to the patients and their family members for their participation in our study. We thank J.P. Adelman for intense discussions and critical reading of the manuscript, A. Haupt for help with bioinformatics and N. Casals for providing CPT1c knockout brains and antibodies targeting CPT1c. This work was supported by grants of the National Research Agency (ANR-10-IAHU-01) and the Fondation pour la Recherche Médicale (DEQ20120323702) to L.C. and by grants of the DFG to R.A.J. (AB393/1-2 and AB393/2-2) and to B.F. (SFB 746, TP 16 and Fa 332/9-1).

## Author contributions

J.S., U.S., L.C., R.A.J. and B.F. conceived the project. All authors performed experiments and analysed data. R.A.J., L.C. and B.F. wrote the manuscript with support from all authors.

## Additional information

**Competing interests:** The authors declare no competing financial interests.

