## [Peer Review File · Nature Communications]

Reviewers' expertise:

Reviewer #1: Synapse proteomics and biochemistry;

Reviewer #2: Structure and function of glutamate receptors;

Reviewer #3: Glutamate receptor trafficking, biochemistry;

Reviewer #4: Genetics of neurodevelopmental disorders.

Reviewers' comments:

Reviewer #1 (Remarks to the Author):

Brechet et al. manuscript follows on from some excellent biochemical work from the laboratory of Bernd Fakler, who have identified several novel and important auxiliary subunits and characterise d their physiological function.

Using QconCAT MS, they take constituents of the AMPAR proteome and perform the reciprocal affinity purification. This shows two distinct populations of AMPAR receptors. In particular, there is one that contains FRRS1l, CPT1c, SAC1, ABHD-6, ABHD-12, PORCN that lacks other non-channel subunits. Using serial immunodepletion-affinity purification, Brechet shows that CPT1c and SAC1 can bind FRRS1l in complexes that lack AMPARs. Brechet shows by MS quantification of AP samples that the abundance of FRRS1l bound to AMPARs is reduced in CPT1c knockout mice over wild type. Next, they identify mutations in consanguineous families with intellectual disability. Next, they use heterologous expression in transformed HEK cells to show that CPT1 and FRRS1l assemble cooperatively with GluA1/2. They also show cooperativity of Sac1 binding to GluA1/2 if CPT1 and FRRS1l are also expressed. Focusing on FRRS1l and CPT1c Brechet use confocal microscopy to show that CPT1c is in the ER and that some FRRS1l colocalizes with CPT1c but not with the mitochondrial homologue, CPT1a. They show that the FRRS1l antibody detects two bands below 37 kDa, whereas recombinant FRRS1l appears as a single lower band when heterologously expressed. Placing a C-terminal tag on the FRRS1l generates a larger band and the lower band. This suggests FRRS1l is proteolytically digested at its C-terminus. Brechet confirm this by MS and identify that the cleavage site (S317) is posttranslationally modified with a GPI anchor. Next, they show FRRS1l by Co-IP that c-terminal truncation of FRRS1l prevents binding to GluA1/2 and that FRRS1l familial intellectual disability mutations reduced apparent expression of FRRS1l and blocked assembly with GluA1/2. They authors now turn to using IF in primary neurons to show that FRRS1l partially co-localizes with calnexin but not vGlut1. Immunogold of rat brain shows FRRS1l in the ER. Finally, they use lentiviral driven knockdown of FRRS1l and electrophysiological recording in brain slices to show that FRRS1l is necessary for normal AMPAR currents. FRRS1l knockdown appears to cause a modest reduction in EPSC suggesting a reduced number of functional synaptic AMPARs. They authors also showed that extrasynaptic AMPAR responses were also reduced following FRRS1l knockdown.

Overall, this is an intriguing and thorough pieces of research, using a remarkably diverse array of methods. The evidence of the importance of FRRS1l for AMPAR receptor biogenesis is important and will be of general interest. Brechet showing the apparent deleterious effect of the intellectual disability mutations is also very important.

There are two main strengths of the manuscript. One is in documenting the association of these proteins in complexes and the other is in providing genetic and knockdown evidence that they are involved with synaptic transmission and behavior. Although the authors provide convincing evidence that the proteins are associated with AMPA receptor in complexes, we don't have any insight into the nature of the interaction/binding (structural requirements, domains, AMPA subunits etc). Nor do we understand how they might work. What might the carnitine 0-palmitoyltransferase do? Does it require

an enzymatic function or is it a scaffold function? And what does the Ferric Chelate Reductase like protein actually do?. The 'functional' experiments don't really tell us much other than FRRS1I and CPT1c are required to produce the AMPA receptor at the membrane. So overall, I find that I am left with a rather superficial insight into the function of FRRS1I and CPT1c and their interaction with AMPA receptors.

Issues to be addressed:

1. Statistics. There are no statistics for most data and figures, except for physiology. This needs to be corrected throughout and significant differences reported.
2. Data deposition. The proteomic data should be provided in supplementary tables and deposited in public domains. See next comment.
3. Figure 1a. Shows a heat map of protein abundance from multiple APs. It is not clear in the figure legend or methods how the numbers were generated to produce this heat map. The units (mol. Abundance [log a.u.]) require clear explanation.
4. Figure 1a,b. Why were not the same assays performed with antibodies to CPT1c? This appears incomplete and the data would be valuable for their hypothesis about the mutual localization and interactions.
5. Figure 2. Refers to 4 different anti- FRRS1I proteins and two different anti- CPT1c antibodies and presents mean +- SEM of 5 and three Aps. This is confusing: what do the error mean? How did the different antibodies perform? The key with red/blue is also unclear.
6. The abstract says "FRRS1I and CPT1c ...specifically and cooperatively bind to the GluA1-4 proteins of AMPA receptor". Can they clarify if this protein is really specific to AMPA receptors or is it that the AMPA receptor is the major binding partner in the brain. Expression databases indicate that FRRS1I and CPT1c are expressed in many non-neuronal tissues where there are not thought to be functional AMPA receptors. Could it be that the FRRS1I and CPT1c are more general rather than AMPAR-specific? Further to this point see the next two comments.
7. They human phenotypes. While it is possible that the phenotypes are due to altered expression of AMPA receptors, I would advise caution in how this is explained (until mouse genetic experiments show some related phenotypes with electrophysiology and rescue). The FRRS1I protein can have other functions that could lead to the intellectual disability phenotype and the authors have not separated these possibilities.
8. Figure 3a. Please label the genotype of all family members in each pedigree. Are all affected individuals homozygous for the FRRS1I mutation? Are any unaffected individuals homozygous for any of the FRRS1I mutations?
9. Figure 4a. Sac1 immunoblot of anti-GluA AP samples (second panel from left). This immunoblot looks under-exposed compared to the other anti-GluA AP panels. The Sac1 band in the GluA1/2-CPT1/FRRS1I lane is faint. There are even fainter bands in the GluA1/2-Sac1 lane and the GluA1/2-Sac1-FRRS1I lane.
10. Page 5. Sentence beginning "Interestingly, these AMPAR subunits displayed..." is confusing. Please re-write.

11. Page 7. First two lines. "And suggest the importance of FRRS1I-containing AMPARs for normal brain development and function". Brechet shows that FRRS1I is found in at least two distinct complexes: a complex that lacks AMPARs and complex that contains AMPARs. It is not known if the mutations in FRRS1I affect the AMPAR containing or the AMPAR-lacking complexes? I recommend removing this statement from the results section.

12. Page 7. Paragraph beginning "Further analysis by confocal...". The authors suggest the confocal data in figure 7b shows CPT1c expression causes a redistribution of FRRS1I from the surface to an intracellular membrane compartment. None of the confocal images show what area of the image is intra- versus extra-cellular. I suggest that if the authors wish to keep that point in the results section text, they probably need to show some surface stain (perhaps using surface biotinylation) or an intracellular marker (perhaps calnexin could be used as Brechet et al us this in Supp. Fig. 3.) to demonstrate that FRRS1I has undergone a redistribution when CPT1c is co-expressed. It would also be helpful to show some evidence of quantitative image analysis to convince the reader of this point.

13. Figure 4c, left. Immunoblot. This is annotated on the left side with two black and one red line. Please describe these annotations in the figure legend.

14. Figure 4c, left. Immunoblot. Please describe what is the "FRRS1IC-tag" in the third lane from the left in the figure legend.

15. The results section on the data in Figures 2 and 4 were quite difficult to follow, especially in contrast to other parts of the text, which were very clear and concise.

16. Figure 5a,b. Why were not the same assays performed for CPT1c? This appears incomplete and the data would be valuable for their hypothesis about the mutual localization and interactions.

Other edits:

1. They should spell out the full name of FRRS1I and CPT1c and describe them in some more detail in the introduction, including information on their expression in other tissues etc.

2. They repeatedly refer to "profound" effects and findings. For this reviewer, none of these findings justify the use of this adjective. Many of their "profound" findings would be better described as expected, or in agreement with, or significant. I also note that some of these "profound" findings are not backed up with statistics and thus it remains for the authors to demonstrate that they are significant.

Reviewer #2 (Remarks to the Author):

In this study Fakler and colleagues study AMPAR complexes associated with FRRS1I and CPT1c, constituents that they identified in an earlier proteomics screen (Schwenk at al., Neuron 2012). They show that AMPARs associated with these components lack the 'classical' core auxiliary subunits and provide some evidence that they constitute early intermediates localizing to the ER. This study is well conducted and will clearly be of interest to the iGluR community. Although the study examines FRSS1I disease mutations it falls short of discussing/investigating mechanistic details of FRRSS1 action and ignores current literature on AMPAR biogenesis, which should be considered in this paper.

The experiments provide an interesting initial characterisation of these AMPAR associated proteins, but the investigation into AMPAR/FRRS1I/CPT1c function currently lacks any development of a mechanism

beyond a "trafficking role". It would be of considerable interest to see at what stage in AMPAR biogenesis these proteins work, do they play a role in initial protein folding, and/or in the assembly of dimers or tetramers, or even during core subunit heteromerisation? Would QconCAT allow them to assess whether FRRS1/CPT1c indeed associates with AMPAR tetramers or could they selectively complex with assembly intermediates (monomers/dimers)?

AMPAR processing in the ER has been investigated and this information should be integrated into the current study; e.g. individual core subunits traffic differently, e.g. GluA3 traffics poorly compared to the other subunits (Coleman et al. 2010); RNA editing and alternative splicing of subunits determines ER exit rates (e.g. Coleman et al. JBC 2010, Penn et al. EMBOJ 2008) – do the current findings bear any relevance to this and might it play a role in selective action of FRRS1/CPT1c.

For example, it appears from Figure 2a that GluA2 is a major component of FRRS1/CPT1c ER complexes consistent with GluA2 retention in the ER due to Q/R editing (Greger et al. Neuron 2002). Is this the case? Also, have they probed blots with anti-GluA2 following anti-GluA1 AP to assess the proportion of AMPAR heteromers? One may expect that ER complexes are incompletely assembled AMPARs as 1Q/2R heteromers are export-competent.

Related to the text on p7 (middle), why is 'data not shown' for the ability CPT1c to retain GluA1 in the ER via interaction with FRRS1? What happens in the presence of GluA2? One might expect FRRS1/CPT1c to drive assembly of GluA1 with GluA2. Likewise, the neuronal results (Figure 2a) show that all FRRS1/CPT1c complexes are associated with GluA2 while being evenly distributed between GluA1 and GluA3, allowing them to recruit either subunit to unassembled GluA2.

The immunocytochemistry in Figs 4b and 5a does not include AMPA staining, GluA2 should be a nice marker for ER-localized AMPARs and should have been included

Do sh-FRRS1 and sh-CPT1c affect rectification of evoked EPSCs? Loss of FRRS1/CPT1c complexes may enable forward trafficking of GluA2-lacking receptors to the synapse before they have the chance to reassemble with GluA2. The loss of 60% is reminiscent of conditional GluA2 knockout (Lu et al., Neuron 2009).

Minors

Figure 1a: is the first red box intentionally selecting CNIH-3 but not CNIH-2? It appears that these two proteins are found to similar levels in these samples. It also appears that there is a significant amount in the ER complexes unlike the TARPs and other core subunits.

P4: 'FRRS1 effectively associates with CPT1c and Sac1 independent of GluA1-4' is not completely true as there is more of these proteins in the AMPAR-containing samples (comparing brown bars against red bars in figure 1b).

P5: 'effective (close to stoichiometric) co-assembly' of FRRS1 and CPT1c is somewhat an overstatement as although every FRRS1 interacts with one CPT1c on average (top panel of Figure 2a), there is an additional population of CPT1c that does not interact with FRRS1 (bottom panel). Accordingly, CPT1c shows an FRRS1-like pattern of preferences for GluA subunits but with reduced amounts.

P5 again: One would also argue that SAC1 and PORCN are barely associated with the FRRS1- or CPT1c-containing AMPARs based on Figure 2a. Figure 4a also shows limited pull-down of SAC1 by AMPARs although it is pulled down by FRRS1. The mention of these is probably fine however as they are not followed up later on.

P10: The sentence starting 'the decrease and increase in EPSC amplitude observed with sh-FRRS1l and sh-CPT1c' is confusing because FRRS1l overexpression, which causes the increase, is not mentioned.

P11: 'Knockdown or exogenous (over-)expression of FRRS1l or CPT1c' suggests both manipulations were applied to both genes/proteins but no data is shown for CPT1c overexpression. Please change to make this clear.

P12: CPT1c knockout mice are mentioned in relation to physiological relevance but the phenotype is not stated. Please briefly summarise. More generally, I'd be nice if the authors could briefly introduce what's currently known about FRRS1 and CPT1c function.

P17 and Figure 8b: there is no evidence that AMPARs assembling with CPT1c and FRRS1l are tetramers.

Supplementary table 2 is missing.

Reviewer #3 (Remarks to the Author):

This is an interesting and generally well-done/convincing study showing that FRRS1i and CPT1c assemble with the pore-forming subunits of AMPA receptors (GluA1-4) in a protein complex distinct from that of the auxiliary subunits including TARPs, cornichons and GSG1L. The authors also show that perturbations of FRRS1L impact synaptic signaling by AMPA receptors. As mutations in FRRS1L cause severe defects in intellectual ability and epilepsy, this work has relevance for both basic and clinical sciences.

The authors first perform proteomic analyses to identify distinct AMPAR assemblies. This work is rigorous, interesting and convincing. The authors next characterize families in which mutations in FRRS1L cause severe intellectual ability. This work is also well done and contributes to an already existing genetic literature concerning mutations in FRRS1L and profound neurodevelopmental disabilities. The authors go on to show that AMPA receptors associated with FRRS1i and CPT1c localize to the ER and that assembly of this complex is disrupted by disease mutations. These are important findings that provide mechanistic insight regarding the pathogenesis of FRRS1i-linked human disease. Finally the authors study the functional effects on AMPA receptors of manipulating FRRS1i and CPT1c protein levels. This work shows that knockdown of FRRS1i or CPT1c reduces the amplitude but not the kinetics of AMPAR-mediated currents. Surprisingly, knockdown of CNIH-2, TARP-2 or TARP-8 had no effect on AMPAR current amplitude. These latter results conflict with previous publications. Also, I could not find in either the main text or figure 7 legend any mention of the data in figure 7b concerning sh-CNIH-2, sh-TARP-2 or sh-TARP-8. This issue must be addressed in detail.

I have additional specific comments below.

1. The immunofluorescence and EM data in figures 4b and 5 must be objectively quantitated.
2. The AMPA/NMDA receptor ratios referred to on page 10 are crucial controls and the traces should be shown in the main figures.
3. The methodology for collection the interpretation of the correlations in figure 8a were not clear to me. For example, do the authors find it interesting that GluA3 and TARP-8 show correlation with FRRS1i but GluA2 and TARP-2 do not?

Reviewer #4 (Remarks to the Author):

This paper describes identification of AMPA-type glutamate receptor (AMPA) complexes that transiently form in the endoplasmic reticulum (ER) through proteomic approaches. The authors describe FRRS1l and CPT1c as critical components of these complexes and show that they cooperatively bind to the GluA1-4 proteins. They also describe three families with intellectual disability and epilepsy and show that biallelic mutations of the FRRS1L gene are responsible for their condition. Finally, the authors show that virus-mediated suppression or overexpression of FRRS1l leads to alteration of synaptic transmission by changing the number of AMPARs in synapses and extra-synaptic sites.

This is a generally well-designed and well-conducted study. The manuscript is concisely written and the data are clearly presented with well-made figures. The strengths of this paper include meticulously conducted proteomic and electrophysiological studies, which beautifully highlight the roles of FRRS1l and CPT1c in AMPAR biogenesis. On the other hand, there are some weaknesses. Mutations in FRRS1L have already reported in two papers as references by the authors (Madeo et al. [Reference #27] and Shaheen et al. [Reference #28]) and therefore not particularly novel. In terms of biological mechanisms of AMPAR regulation, how FRRS1l and CPT1c controls the number of AMPAR in synapses and extra-synaptic sites is not explored.

I have following comments:

1] The authors report three additional families with intellectual disability and epilepsy due to FRRS1L mutations. However, this clinical part of the study appears to add few new clinical insights, beyond what have already been reported previously by Madeo et al. and Shaheen et al. Did the authors find any genotype-phenotype correlation? For example, the authors state that "...the K155E substitution mutant was still able to interact with GluA1/A2, albeit less effectively than WT FRRS1l" (page 8). Did the individuals with the K155E mutation have milder clinical phenotype than the ones with truncating mutations? Madeo et al. reports choreoathetosis as a cardinal feature of FRRS1L mutations. Did any of the affected individuals reported herein have movement disorder? On another note, do the biological findings presented herein inform about pathogenesis and potential intervention for the condition?

2] The effect of FRRS1L knockdown on AMPAR-mediated current has previously been shown by Madeo et al., though the current manuscript includes much more detailed electrophysiological studies. What is not known is how FRRS1l regulates the AMPARs. The authors state that this is to be elucidated in the future (page 7), but additional data on molecular effects of FRRS1l (or mutation thereof) would be desirable. For example, what happens to GluA tetramers in ER if there is no FRRS1l (or CPT1c)? Do they remain in ER or leave ER but not get trafficked correctly to synapses?

3] As a minor point, the following statement is not entirely clear: "...all membrane proteins with different topology and suggested localization to the ER..." (page 5). How was ER localization of FRRS1l and CPT1c suspected?

Responses to the reviewers' comments

Reviewer 1

We thank the reviewer for the positive comments on our work and his suggestions for further improvement that were incorporated into the revised manuscript.

General comment(s):

Overall, this is an intriguing and thorough piece of research, using a remarkably diverse array of methods. The evidence of the importance of FRRS1I for AMPAR receptor biogenesis is important and will be of general interest. Brechet showing the apparent deleterious effect of the intellectual disability mutations is also very important.

There are two main strengths of the manuscript. One is in documenting the association of these proteins in complexes and the other is in providing genetic and knockdown evidence that they are involved with synaptic transmission and behavior. Although the authors provide convincing evidence that the proteins are associated with AMPA receptor in complexes, we don't have any insight into the nature of the interaction/binding (structural requirements, domains, AMPA subunits etc). Nor do we understand how they might work. What might the carnitine 0-palmitoyltransferase do? Does it require an enzymatic function or is it a scaffold function? And what does the Ferric Chelate Reductase like protein actually do? The 'functional' experiments don't really tell us much other than FRRS1I and CPT1c are required to produce the AMPA receptor at the membrane. So overall, I find that I am left with a rather superficial insight into the function of FRRS1I and CPT1c and their interaction with AMPA receptors.

While we appreciate the positive judgement on the overall concept and validity of our work, we firmly disagree with the statement that the ms does not provide insights into 'the nature of the interaction' (between GluAs and FRRS1I) and into its mode of operation (suggested involvement of potential enzymatic activities).

Thus:

- (a) The structural requirements of the interaction have in fact been investigated in detail and the hydrophobic domain in the C-terminus of FRRS1I (per se or by guiding GPI-anchoring at S317) has been identified as the decisive domain for interaction (Figure 4c, d); deletion of this domain by two disease mutations (Q321*. V195*) abolishes FRRS1I binding (Fig. 4d) to GluAs (occurring at sites different from the binding pockets of CNIH/TARPs, see Schwenk et al., 2012).
- (b) Potential 'enzymatic activities' as inferred from the protein names are not involved in the action of FRRS1I/CPT1c.

For FRRS1I, the database name is largely misleading (see note of CAUTION given in UniProtKB: *Named FRRS1I by HGNC because it shares limited sequence similarity with FRRS1. However, sequence similarities lie outside of the reductase region, suggesting it has no oxidoreductase activity*).

For CPT1c, published work by the Casals group indicated that enzymatic activity (in particular (de)-palmitoylation) is not involved in CPT1c-mediated changes in AMPAR expression (e.g. Gratacos-Batlle et al, 2014), and other papers (e.g. Wolfgang et al, 2006; Price et al, 2002) find no enzymatic activity for CPT1c at all (in contrast to CPT1a/b).

Specific issues

1. Statistics. There are no statistics for most data and figures, except for physiology. This needs to be corrected throughout and significant differences reported.

Statistics have been included into the ms where meaningful. In particular, we have added statistics for the proteomic data in Figures 1, 2 and 8 (as requested by the reviewer). The data on the two-step APs (Figure 1b) containing combined data for rats and mice (to show the close resemblance) in the original ms, were separated in the revised ms (rat data in Fig. 1b, mouse data in Supplementary Figure 1b); EM data were quantified (evaluation of 848 immunoparticles).

Data on genetics, as well as on all functional measurements had already been indicated in the original ms.

2. Data deposition. The proteomic data should be provided in supplementary tables and deposited in public domains. See next comment.

3. Figure 1a. Shows a heat map of protein abundance from multiple APs. It is not clear in the figure legend or methods how the numbers were generated to produce this heat map. The units (mol. Abundance [log a.u.]) require clear explanation.

Generation of the heat map (Figure 1a) is detailed in the Methods section and has been introduced in previous work (in particular, the QConCAT technique for absolute quantification, see also Schwenk et al., Neuron 2012).

The data from proteomic analyses have been summarized and added to the revised ms as Supplementary Table 3; the original raw data are deposited to public databases as suggested by the reviewer.

4. Figure 1a,b. Why were not the same assays performed with antibodies to CPT1c? This appears incomplete and the data would be valuable for their hypothesis about the mutual localization and interactions.

Experiments as in Figures 1a, b were not performed for CPT1c, as the *anti-CPT1c* antibodies (ABs) in hand did not allow for depleting affinity-purifications (APs) thus precluding clear separation of AMPAR pools.

5. Figure 2. Refers to 4 different anti- FRRS1l proteins and two different anti- CPT1c antibodies and presents mean +- SEM of 5 and three Aps. This is confusing: what do the error mean? How did the different antibodies perform? The key with red/blue is also unclear.

The discrepancy results from one AP in either case using a mixture of the different ABs; red bars refer to the respective AB target protein in either of the two APs (FRRS1l (upper panel), CPT1c (lower panel)) that were used for normalization (of protein amounts). This has been clarified in the revised legend to Figure 2a.

6. The abstract says “FRRS1l and CPT1c ...specifically and cooperatively bind to the GluA1-4 proteins of AMPA receptor”. Can they clarify if this protein is really specific to AMPA receptors or is it that the AMPA receptor is the major binding partner in the brain. Expression databases indicate that FRRS1l and CPT1c are expressed in many non-neuronal tissues where there are not thought to be functional AMPA receptors. Could it be that the FRRS1l and CPT1c are more general rather than AMPAR-specific? Further to this point see the next two comments.

Searching public databases (including EMBL-EBI Expression Atlas, Human Proteome Map, Human Protein Atlas) indicated that the only tissue displaying consistent and high expression of both FRRS1I and CPT1c is the brain (adult, less in fetal brain). Moreover, several papers report focused expression in brain (e.g. Price et al, 2002; Madeo et al, 2016); in fact, the only non-neuronal tissue displaying some expression outside the brain was testis.

In this context, the statement of FRRS1I/CPT1c being specific interactors of AMPARs appears justified, in particular as there is no indication for FRRS1I and CPT1c being 'more general'.

7. They human phenotypes. While it is possible that the phenotypes are due to altered expression of AMPA receptors, I would advise caution in how this is explained (until mouse genetic experiments show some related phenotypes with electrophysiology and rescue). The FRRS1I protein can have other functions that could lead to the intellectual disability phenotype and the authors have not separated these possibilities.

Detailed proteomic analyses (multi-epitope APs with target-knockout controls, Figure 2a) revealed that the vast majority of interaction partners - identified for both (!) FRRS1I and CPT1c - are AMPAR constituents (with protein amounts totaling to more than 99% of all partner proteins identified). Based on these findings and the aforementioned observations (see point 6), it appears rather unlikely that FRRS1I is serving a function that is both independent from AMPARs and relevant for the presented disease.

8. Figure 3a. Please label the genotype of all family members in each pedigree. Are all affected individuals homozygous for the FRRS1I mutation? Are any unaffected individuals homozygous for any of the FRRS1I mutations?

The disease mutations reported here were identified in two independent systematic studies analyzing cohorts of consanguineous families for genes causative for autosomal recessive intellectual disability. In line with a 'recessive disorder' all affected patients are homozygous for the respective mutation, while the healthy siblings and parents are either heterozygous for the mutation or homozygous for the wildtype allele.

Genotypes have been added to the revised Figure 3a, as suggested by the reviewer.

9. Figure 4a. Sac1 immunoblot of anti-GluA AP samples (second panel from left). This immunoblot looks under-exposed compared to the other anti-GluA AP panels. The Sac1 band in the GluA1/2-CPT1/FRRS1I lane is faint. There are even fainter bands in the GluA1/2-Sac1 lane and the GluA1/2-Sac1-FRRS1I lane

Equivalent amounts of input and eluates of all APs were loaded, separated (on the same gel) and Western-probed under the same conditions to guarantee comparability (and avoid signal distortion by over-exposure). Although faint, the dynamic range of the Sac1-staining clearly indicated more effective binding to GluAs in the presence of both FRRS1I and CPT1c (over the individual co-assemblies). This result from heterologous expression is perfectly in line with the AP results from native tissue that show Sac1 as a specific but low abundant partner of AMPAR assemblies (Figures 2a, b)

Page 5. Sentence beginning “Interestingly, these AMPAR subunits displayed...” is confusing. Please re-write.

The sentence was re-phrased in the revised ms.

11. Page 7. First two lines. “And suggest the importance of FRRS1I-containing AMPARs for normal brain development and function”. Brechet shows that FRRS1I is found in at least two distinct complexes: a complex that lacks AMPARs and complex that contains AMPARs. It is not known if the mutations in FRRS1I affect the AMPAR containing or the AMPAR-lacking complexes? I recommend removing this statement from the results section.

As pointed out above (point 7), AMPARs are (by far) the predominant interactors of FRRS1I and it appears therefore reasonable to assume that the entirety of FRRS1I protein is present in an equilibrium between AMPAR-associated and AMPAR-free pools. Moreover, all mutations are finally shown to affect the assembly of FRRS1I with AMPARs. In this sense, we would prefer to leave the sentence as it stands.

12. Page 7. Paragraph beginning “Further analysis by confocal...”. The authors suggest the confocal data in figure 7b shows CPT1c expression causes a redistribution of FRRS1I from the surface to an intracellular membrane compartment. None of the confocal images show what area of the image is intra- versus extra-cellular. I suggest that if the authors wish to keep that point in the results section text, they probably need to show some surface stain (perhaps using surface biotinylation) or an intracellular marker (perhaps calnexin could be used as Brechet et al us this in Supp. Fig. 3.) to demonstrate that FRRS1I has undergone a redistribution when CPT1c is co-expressed. It would also be helpful to show some evidence of quantitative image analysis to convince the reader of this point.

Re-distribution of FRRS1I (from plasma membrane to ER) by the ER-resident CPT1c is shown in Figure 4b, in direct contrast to the failure (to induce re-distribution) by CPT1a.

To confirm this re-distribution, we have added two additional experiments: A surface biotinylation assay (as suggested by the reviewer) and electrophysiological recordings. These results were included into the revised ms as new Supplementary Figure 4.

13. Figure 4c, left. Immunoblot. This is annotated on the left side with two black and one red line. Please describe these annotations in the figure legend.

14. Figure 4c, left. Immunoblot. Please describe what is the “FRRS1IC-tag” in the third lane from the left in the figure legend.

The color-coding of the distinct MW bands of the FRRS1I protein, as well as the C-terminal tag used were added to the revised caption of Figure 4c (and had been indicated in the text, supplementary figure and methods part of the original ms).

15. The results section on the data in Figures 2 and 4 were quite difficult to follow, especially in contrast to other parts of the text, which were very clear and concise.

We tried to elaborate on this as far as possible and in line with the reports and requests from the other reviewers.

16. *Figure 5a,b. Why were not the same assays performed for CPT1c? This appears incomplete and the data would be valuable for their hypothesis about the mutual localization and interactions.*

These experiments were not possible due to the unfavorable properties of the *anti-CPT1c* ABs in hand.

Other edits:

1. *They should spell out the full name of FRRS1l and CPT1c and describe them in some more detail in the introduction, including information on their expression in other tissues etc.*

The names of both proteins were explicitly given in the introduction section (although, these names are misleading with respect to their (primary) function, see response to the general comment).

2. *They repeatedly refer to “profound” effects and findings. For this reviewer, none of these findings justify the use of this adjective. Many of their “profound” findings would be better described as expected, or in agreement with, or significant. I also note that some of these “profound” findings are not backed up with statistics and thus it remains for the authors to demonstrate that they are significant.*

This word was replaced for most parts.

Reviewer 2

We appreciate the reviewers' positive comments, as well as his criticism that prompted additional experiments included into the revised manuscript as detailed below in the responses to the reviewer's comments.

Specific comments:

1. The experiments provide an interesting initial characterisation of these AMPAR associated proteins, but the investigation into AMPAR/FRRS1/CPT1c function currently lacks any development of a mechanism beyond a "trafficking role". It would be of considerable interest to see at what stage in AMPAR biogenesis these proteins work, do they play a role in initial protein folding, and/or in the assembly of dimers or tetramers, or even during core subunit heteromerisation? Would QconCAT allow them to assess whether FRRS1/CPT1c indeed associates with AMPAR tetramers or could they selectively complex with assembly intermediates (monomers/dimers)?

As requested by the reviewer, we have added data on native gel-separations of AMPARs and their evaluation (for subunit composition) by both Western-blot and cryo-slicing BN-MS (using QconCAT for quantification). These data demonstrate that FRRS1/CPT1c do only form complexes with GluA-tetramers, association with assembly intermediates were not detected in the membrane fractions from rodent brain (Supplementary Figure 5b, c). Consequently, FRRS1/CPT1c are not expected to influence the folding and assembly of the AMPAR pores.

Moreover, we probed interaction of FRRS1/CPT1c with GluA1-4 (separately) for potential preferences in assembly. The results of respective *anti-FRRS1* APs, however, did not provide any indication for differences in FRRS1-binding to the individual GluA proteins (Supplementary Figure 5a).

Importantly, FRRS1 did not show any preference for GluA2 (used in R-edited form here).

These additional data were added to the revised ms as Supplementary Figure 5.

AMPA processing in the ER has been investigated and this information should be integrated into the current study; e.g. individual core subunits traffic differently, e.g. GluA3 traffics poorly compared to the other subunits (Coleman et al. 2010); RNA editing and alternative splicing of subunits determines ER exit rates (e.g. Coleman et al. JBC 2010, Penn et al. EMBOJ 2008) – do the current findings bear any relevance to this and might it play a role in selective action of FRRS1/CPT1c.

ER processing has been mostly investigated in heterologous expression experiments (e.g. Coleman et al) with appearance of functional receptors at the plasma membrane or mature glycosylation pattern as a read-out. While the relevance of these experiments (missing the essential ER components identified here) is not immediately clear in the context of this work, factors influencing ER-exit have not been studied in this ms, but may contribute to the 'trafficking' of AMPARs to the surface. Importantly though, these factors do not compensate for disturbances introduced into biogenesis by knock-down of FRRS1 and/or CPT1c.

Both facts have been stated in the Discussion section of the revised ms (p. 13, lines 28-35); the suggested citations were added.

For example, it appears from Figure 2a that GluA2 is a major component of FRRS1/CPT1c ER complexes consistent with GluA2 retention in the ER due to Q/R editing (Greger et al. Neuron 2002). Is this the case? Also, have they probed blots with anti-GluA2 following anti-GluA1 AP to assess the proportion of AMPAR heteromers? One may expect that ER complexes are incompletely assembled AMPARs as 1Q/2R heteromers are export-competent.

As detailed in the new Supplementary Figure 5 (and also in Schwenk et al, 2012), GluA2 is the most abundant pore-forming subunit of AMPAR assemblies both in their 'plasma membrane' form (low molecular mass range, assembly with CNIH/TARPs), as well as in their FRRS1/CPT1c-associated 'ER-form' (high molecular mass range, at the maximum of FRRS1/CPT1c). In fact, there is no observable difference between plasma membrane- and FRRS1-associated AMPARs with respect to the GluA-composition.

The proportion of GluA heteromers has been detailed in previous work by AB-shift assays with anti-GluA1 and anti-GluA2 ABs showing that the vast majority of detectable AMPARs (in all membranes) are heteromers (Schwenk et al, 2012; Figure 3).

Related to the text on p7 (middle), why is 'data not shown' for the ability CPT1c to retain GluA1 in the ER via interaction with FRRS1? What happens in the presence of GluA2? One might expect FRRS1/CPT1c to drive assembly of GluA1 with GluA2. Likewise, the neuronal results (Figure 2a) show that all FRRS1/CPT1c complexes are associated with GluA2 while being evenly distributed between GluA1 and GluA3, allowing them to recruit either subunit to unassembled GluA2.

There appears to be some misunderstanding here:

The data (from biotinylation assays and patch-clamp recordings) demonstrating redistribution of GluA1-associated FRRS1 by CPT1c (cited as not shown in the original ms on p.7) were added to the revised ms as new Supplementary Figure 4.

There is no effective ER-retention of GluA tetramers as stated by reviewer, rather transient binding of FRRS1/CPT1c to GluA tetramers catalyses assembly of the latter with CNIH/TARP thus rendering them competent for ER-exit and transport to the plasma membrane (see Figure8b).

The immunocytochemistry in Figs 4b and 5a does not include AMPA staining, GluA2 should be a nice marker for ER-localized AMPARs and should have been included.

The goal of Figure 5a was probing absence of FRRS1 from the synaptic compartment in brain neurons (contained in hippocampal slice preparations), as suggested by the ER-staining obtained in culture cells. GluA2 (showing both, synaptic, extrasynaptic and intracellular staining) would not have been a suited marker for these experiments. Nonetheless, we added experiments with heterologously expressed GluA2(R) to the revised ms (new Supplementary Figure 3b) that show close co-localization of GluA2 and FRRS1/CPT1c in the ER.

Do sh-FRRS1 and sh-CPT1c affect rectification of evoked EPSCs? Loss of FRRS1/CPT1c complexes may enable forward trafficking of GluA2-lacking receptors to the synapse before they have the chance to reassemble with GluA2. The loss of 60% is reminiscent of conditional GluA2 knockout (Lu et al., Neuron 2009).

Rectification was not tested, but the gating characteristics (also suitable to reflect the respective changes) did not reveal any differences (see Figure 6, 7); a reduction to (not of) 60% may not be a strong indicator in this sense.

Minor comments

Figure 1a: is the first red box intentionally selecting CNIH-3 but not CNIH-2? It appears that these two proteins are found to similar levels in these samples. It also appears that there is a significant amount in the ER complexes unlike the TARPs and other core subunits.

The framing of the red box in the heat map (with a dynamic range of 5 orders of magnitude) for CNIH2/3 was somewhat empirical and drawn to be consistent with the AP results shown in Figure 2a – where CNIH2 was co-purified consistently, albeit at very small amounts (less than 1%) with ER-resident (FRRS1I-containing) AMPARs. This small amount may reflect the transitional priming complex forming during biogenesis (Figure 8b).

P4: 'FRRS1I effectively associates with CPT1c and Sac1 independent of GluA1-4' is not completely true as there is more of these proteins in the AMPAR-containing samples (comparing brown bars against red bars in figure 1b).

This sentence was meant to emphasize robust assembly even in the absence of GluAs, rather than making a comparative statement on the equilibrium between GluA-free and GluA-associated CPT1c or Sac1 (see also Figure 4a).

P5: 'effective (close to stoichiometric) co-assembly' of FRRS1I and CPT1c is somewhat an overstatement as although every FRRS1I interacts with one CPT1c on average (top panel of Figure 2a), there is an additional population of CPT1c that does not interact with FRRS1I (bottom panel). Accordingly, CPT1c shows an FRRS1I-like pattern of preferences for GluA subunits but with reduced amounts.

This sentence was changed to emphasize that 'close to stoichiometric' refers to anti-FRRS1I APs only.

P5 again: One would also argue that SAC1 and PORCN are barely associated with the FRRS1I- or CPT1c-containing AMPARs based on Figure 2a. Figure 4a also shows limited pull-down of SAC1 by AMPARs although it is pulled down by FRRS1I. The mention of these is probably fine however as they are not followed up later on.

This was left as in the original ms.

P10: The sentence starting 'the decrease and increase in EPSC amplitude observed with sh-FRRS1I and sh-CPT1c' is confusing because FRRS1I overexpression, which causes the increase, is not mentioned.

This has been rephrased to avoid confusion.

P11: 'Knockdown or exogenous (over-)expression of FRRS1I or CPT1c' suggests both manipulations were applied to both genes/proteins but no data is shown for CPT1c overexpression. Please change to make this clear.

This has been rephrased to avoid confusion.

P12: CPT1c knockout mice are mentioned in relation to physiological relevance but the phenotype is not stated. Please briefly summarise. More generally, I'd be nice if the authors could briefly introduce what's currently known about FRRS1 and CPT1c function.

The information on FRRS1 and CPT1c as available through the current literature has been added to the revised ms (Introduction, Results and Discussion).

P17 and Figure 8b: there is no evidence that AMPARs assembling with CPT1c and FRRS1 are tetramers.

As pointed out above, these data are explicitly shown in the new Supplementary Figure 5 (but have also been shown in Schwenk et al., 2012 (Fig. 2)).

Supplementary table 2 is missing.

We made sure that this table is part of the revised ms.

Reviewer 3

We thank the reviewer for the positive comments on our work and his suggestions for further improvement.

Specific comments:

Surprisingly, knockdown of CNIH-2, TARP-2 or TARP-8 had no effect on AMPAR current amplitude. These latter results conflict with previous publications. Also, I could not find in either the main text or figure 7 legend any mention of the data in figure 7b concerning sh-CNIH-2, sh-TARP-2 or sh-TARP-8. This issue must be addressed in detail.

The data demonstrating lack of effect on EPSC amplitude by sh-CNIH-2 and sh-TARP-2/8 were obtained from hilar mossy cells (MCs) and described in the original ms (p.10, first section); in particular, the result of sh-CNIH-2 was published in previous work, similar to the results on sh-control (see Boudkkazi et al, 2014). The lack of effect for sh-TARP-2/8 may not be so surprising, as the AMPARs in MCs are largely assembled from GluA1/2 and CNIH-2 (Boudkkazi et al, 2014), in contrast to the most often used CA1 pyramidal cells or granule cells that express TARPs 2 and 8, respectively.

This information was added to the revised ms to avoid confusion (p. 10, lines 34-40).

1. The immunofluorescence and EM data in figures 4b and 5 must be objectively quantitated.

The respective quantification has been added to the revised ms as suggested by the reviewer.

(p. 9: More detailed, 96.1% of all immunoparticles evaluated in 29 neurons (815 out of a total of 848 particles) showed ER-localization, while the remaining 3.9% (33 particles) appeared at or next to the plasma membrane. In either case, the staining was mostly observed in the soma or the proximal dendrites, but did not appear in the synaptic compartment (Fig. 5a).

2. The AMPA/NMDA receptor ratios referred to on page 10 are crucial controls and the traces should be shown in the main figures.

As suggested by the reviewer, traces and evaluation have been added to the revised ms as a new Supplementary Figure 9.

3. The methodology for collection the interpretation of the correlations in figure 8a were not clear to me. For example, do the authors find it interesting that GluA3 and TARP-8 show correlation with FRRS1i but GluA2 and TARP-2 do not?

These correlation analyses used the data on AMPAR assemblies determined across brain regions (Schwenk et al., 2014) to probe relevance of FRRS1I for AMPAR assemblies that exhibit marked differences in the subunit composition.

Interestingly, these correlations indicated that FRRS1I displayed highest correlation (close to 1) with the sum of all core subunits (CNIHs, TARPs, GSG1I) over the individual subunits or the sum of all GluAs thus strongly suggesting that all AMPARs at the plasma membrane go through FRRS1I-assemblies in the ER.

Reviewer 4

We thank the reviewer for the positive comments on our work and his suggestions for further improvement.

Specific comments:

...On the other hand, there are some weaknesses. Mutations in FRRS1L have already reported in two papers as referenced by the authors (Madeo et al. [Reference #27] and Shaheen et al. [Reference #28]) and therefore not particularly novel. In terms of biological mechanisms of AMPAR regulation, how FRRS1l and CPT1c controls the number of AMPAR in synapses and extra-synaptic sites is not explored.

Our WES analyses identified two additional mutations (one of them, Q321* (family B), is shared with both the Madeo et al. and the Shaheen et al. papers) as stated in the ms. However, while neither of the two papers investigates the effects of the presented mutation(s) on AMPAR biology in the context of brain neurons, we present biochemical and functional analyses of FRRS1l (and its partner CPT1c) and provide the first presentation of its impact for the cell biology of AMPARs – namely its role in the biogenesis of the receptor channels in the ER.

These analyses finally promoted a model summarizing the mode of operation for FRRS1l/CPT1c. In fact, this model provides a straightforward explanation of how biogenesis finally controls the number of AMPARs at the plasma membrane (in synaptic and extra-synaptic sites).

For making this more readily accessible we have added further data on the biogenesis (see new Supplementary Figures 4, 5 and 9) and revised presentation and description of the model (new Figure 8b).

1] The authors report three additional families with intellectual disability and epilepsy due to FRRS1L mutations. However, this clinical part of the study appears to add few new clinical insights, beyond what have already been reported previously by Madeo et al. and Shaheen et al. Did the authors find any genotype-phenotype correlation? For example, the authors state that "...the K155E substitution mutant was still able to interact with GluA1/A2, albeit less effectively than WT FRRS1l" (page 8). Did the individuals with the K155E mutation have milder clinical phenotype than the ones with truncating mutations? Madeo et al. reports choreoathetosis as a cardinal feature of FRRS1L mutations. Did any of the affected individuals reported herein have movement disorder? On another note, do the biological findings presented herein inform about pathogenesis and potential intervention for the condition?

The phenotype segregating with the K155E mutation (family A) is indeed less severe than the one observed in the two other families. In particular, opposite to what is observed in families B and C, patients from family A showed no sign of muscular hypotonia or neuro-regression. Moreover, while independent walk was never achieved by patients from families B and C, the three affected siblings of family A reached independent walk at ages between 13 and 24 months. The milder phenotype associated with K155E may in fact correlate with the less reduced effect on GluA-FRRS1l/CPT1c association (Figure 4d).

Movement disorder was not observed in any of the affected patients.

All these informations were added to the revised ms (p. 6, last paragraph, p. 7, first paragraph).

2] *The effect of FRRS1L knockdown on AMPAR-mediated current has previously been shown by Madeo et al., though the current manuscript includes much more detailed electrophysiological studies. What is not known is how FRRS1I regulates the AMPARs. The authors state that this is to be elucidated in the future (page 7), but additional data on molecular effects of FRRS1I (or mutation thereof) would be desirable. For example, what happens to GluA tetramers in ER if there is no FRRS1I (or CPT1c)? Do they remain in ER or leave ER but not get trafficked correctly to synapses?*

As outlined above (general comment), our detailed analyses of the cell biology of FRRS1I both *in-vitro* (Figs. 1, 2, 4) and *in-vivo* (Figs. 5-7, 8a), do provide a straightforward explanation of how FRRS1I controls biogenesis of AMPARs (catalyzing assembly of GluA tetramers with CNIHs and TARPs that finally render the AMPAR assemblies competent for ER exit and delivery to the plasma membrane). Accordingly, in the absence of FRRS1I/CPT1c GluA tetramers can still assemble with the auxiliary subunits (as directly shown in the knock-down experiments in Figures 6, 7 and 8a), albeit at lower efficiency thus generating less mature AMPARs.

The cited sentence (p. 12 (not 7) of the original ms) was indeed misleading, as correctly stated by the reviewer. This has been corrected and the molecular mechanism of FRRS1I function (pictured in Figure 8b) has been thoroughly described in the revised ms (p. 13, 2nd paragraph).

3] *As a minor point, the following statement is not entirely clear: "...all membrane proteins with different topology and suggested localization to the ER..." (page 5). How was ER localization of FRRS1I and CPT1c suspected?*

In fact, localization of FRRS1I and CPT1c was not suspected, but rather investigated in detail in this work by EM and fluorescence microscopy for FRRS1I (see Figures 4 and 5), and by fluorescence microscopy (co-staining of CPT1c and the ER-marked calnexin) for CPT1c (see Supplementary Figure 3).

ER-localization of CPT1c has also been reported in the cited literature (Carrasco et al, 2012; Stefan et al, 2011).

REVIEWERS' COMMENTS:

Reviewer #1 (Remarks to the Author):

I am satisfied with almost all the changes in response to the points we made except for point 12, which does not seem sound because:

- 1) No quantification of confocal data (in spite of a request to provide it). The reader is only given one or two cells on which to base a conclusion.
- 2) The added supplementary figure 4 showing a biotinylation assay appears to have not worked in some way. The To (total membrane) lane in the '+FRRS1I' condition appears to have no FRRS1I compared to the 'S' (surface membrane fraction). Unless I have misunderstood the assay, I would have expected the 'To' to have a great abundance than the 'S' fraction.
- 3) The second panel of the supply. Fig4 doesn't appear to add anything (unless I have missed something): I do not understand how recording AMPAR recovery from desensitisation has any direct bearing on the question of whether Cpt1c recruits FRRS1I from the surface to the ER.

Reviewer #2 (Remarks to the Author):

The authors have mostly addressed my concerns - I recommend publication.

Reviewer #3 (Remarks to the Author):

The authors have successfully addressed my concerns. This is an elegant and comprehensive study that is now suitable for publication in Nature Communications.

Reviewer #4 (Remarks to the Author):

This is a revised manuscript. The authors addressed the points raised by the reviewers. There is now a better description of genotype-phenotype correlation. The molecular mechanism of FRRS1I function is described in more detail and this makes it clearer to readers.

Responses to the reviewers' comments

Reviewer 1

We thank the reviewer for the positive comments, the suggestion for further improvement was incorporated into the revised manuscript as outlined below.

I am satisfied with almost all the changes in response to the points we made except for point 12, which does not seem sound because:

1) No quantification of confocal data (in spite of a request to provide it). The reader is only given one or two cells on which to base a conclusion.

2) The added supplementary figure 4 showing a biotinylation assay appears to have not worked in some way. The To (total membrane) lane in the '+FRRS1I' condition appears to have no FRRS1I compared to the 'S' (surface membrane fraction). Unless I have misunderstood the assay, I would have expected the 'To' to have a great abundance than the 'S' fraction.

3) The second panel of the supply. Fig4 doesn't appear to add anything (unless I have missed something): I do not understand how recording AMPAR recovery from desensitisation has any direct bearing on the question of whether Cpt1c recruits FFRS1I from the surface to the ER.

1) The confocal images shown in Fig. 4b were meant to be representative. Notwithstanding, we have added a quantification of the redistribution effect of CPT1c (from plasma membrane to intracellular membranes) to the revised legend of Figure 4b as requested.

It reads:

Note marked re-distribution of FRRS1I upon co-expression of CPT1c from plasma membrane to intracellular membranes (FRRS1I staining: 93% (of 501 cells co-expressing FRRS1I and CPT1c) intracellular only, 6.8% intracellular and plasma membrane, 0.2% plasma membrane only).

2) There appears to be some misunderstanding here. The S (surface) fraction is enriched 12.5-fold compared to the To (total) and In (intracellular) fractions; this is indicated in the Methods section, and is explicitly stated in the revised legend of Supplementary Figure 4a to avoid misunderstanding.

(Additional note: Expression/stability of the FRRS1I protein is increased upon co-expression of CPT1c).

3) The electrophysiological recordings (second panel of Supplementary Figure 4), add, in fact, a strong point to the redistribution of FRRS1I by CPT1c, as they selectively monitor protein in the plasma membrane. Technically, the recordings exploit the 10-fold prolongation in recovery from desensitization induced in GluA1 AMPARs by co-assembled FRRS1I. Co-expression of CPT1c prevented this prolongation (by retaining FRRS1I in the ER), in contrast to the mitochondrial CPT1a which does not bind to FRRS1I and, consequently, not affect the FRRS1I-mediated prolongation in the recovery time constant.

Together, all the three (independent and orthogonal) approaches (microscopy, biotinylation, electrophysiology) indicate that FRRS1I is effectively re-distributed from the plasma membrane to intracellular membrane (ER) compartments.